# Modeling Neural Activity with Conditionally Linear Dynamical Systems

Victor Geadah[1, 2]       Amin Nejatbakhsh[2]       David Lipshutz[2, 3]

Jonathan W. Pillow[1, 4]       Alex H. Williams[2, 5]

[1]Program in Applied and Computational Mathematics, Princeton University, Princeton NJ
[2]Center for Computational Neuroscience, Flatiron Institute, New York City NY
[3]Department of Neuroscience, Baylor College of Medicine, Houston TX
[4]Princeton Neuroscience Institute, Princeton NJ
[5]Center for Neural Science, New York University, New York City NY

## Abstract

Neural population activity exhibits complex, nonlinear dynamics, varying in time, over trials, and across experimental conditions. Here, we develop *Conditionally Linear Dynamical System* (CLDS) models as a general-purpose method to characterize these dynamics. These models use Gaussian Process (GP) priors to capture the nonlinear dependence of circuit dynamics on task and behavioral variables. Conditioned on these covariates, the data is modeled with linear dynamics. This allows for transparent interpretation and tractable Bayesian inference. We find that CLDS models can perform well even in severely data-limited regimes (e.g. one trial per condition) due to their Bayesian formulation and ability to share statistical power across nearby task conditions. In example applications, we apply CLDS to model thalamic neurons that nonlinearly encode heading direction and to model motor cortical neurons during a cued reaching task.

## 1   Introduction

A central problem in neuroscience is to capture how neural dynamics are affected by external sensory stimuli, task variables, and behavioral covariates. To address this, a longstanding line of research has focused on characterizing neural dynamics through recurrent neural networks (RNNs) and their probabilistic counterparts, state-space models (SSMs; for reviews, see [1, 2, 3]).

Early work in this area utilized latent linear dynamical systems (LDS) with Gaussian observation noise. Although these assumptions are restrictive, they are beneficial in two respects. First, they simplify probabilistic inference by enabling Kalman smoothing and expectation maximization (EM)—two classical and highly effective methods [4]. Second, they produce models that are mathematically tractable to analyze with well-established tools from linear systems theory [5]. Indeed, many influential results in theoretical neuroscience have come from purely linear models [6, 7, 8].

In reality, most neural circuits do not behave like time-invariant linear systems. Thus, more recent work from the machine learning community has cataloged a variety of nonlinear models for neural dynamics. Although these new models often predict held out neural data more accurately than LDS models, they are generally more difficult to fit and more difficult to understand. Thus, there has been a proliferation of competing models for neural data analysis, such as RNNs [9], transformers [10, 11], and diffusion-based methods [12], as well as competing inference methods, such as generalized

39th Conference on Neural Information Processing Systems (NeurIPS 2025).

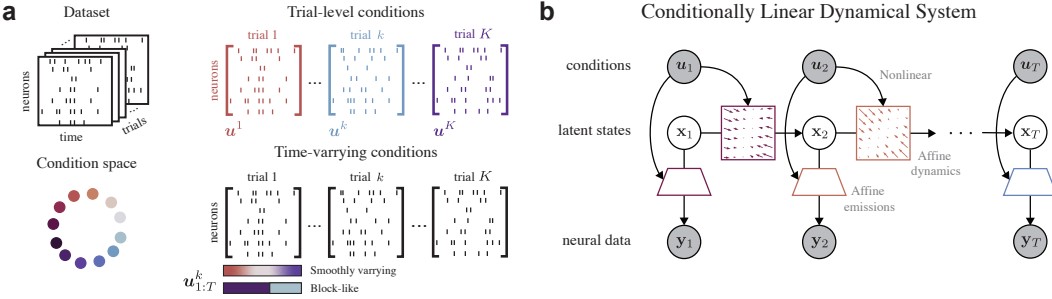

Figure 1: **(a)** Neural dataset consisting of spike trains collected over multiple trials, along with corresponding experimental conditions. **(b)** Conditionally Linear Dynamical Systems are *linear* in state-space dynamics and capture *nonlinear* dependencies over conditions. Shaded nodes are observed, clear nodes are latent.

teacher forcing [13], amortized variational inference [9], and sequential Monte Carlo [14]. Choosing among these strategies and scientifically interpreting the outcomes is challenging.

Here we describe *Conditionally Linear Dynamical Systems* (CLDS) as a framework to jointly capture some of the benefits of the classical (i.e. linear) and contemporary (i.e. nonlinear) approaches to modeling neural data. CLDS models parametrize a collection of LDS models that vary smoothly as a function of an observed variable $\boldsymbol{u}_t$ (e.g. measured sensory input or behavior at time $t$). Assuming the presence of $\boldsymbol{u}_t$ is often a feature—not a bug—of this approach. Indeed, a common goal in neuroscience is to relate measured sensory or behavioral covariates to neural activity. Additional features of the CLDS framework include:

1. CLDS models are locally interpretable. Conditioned on $\boldsymbol{u}_t$, the dynamics are linear and amenable to a number of classical analyses.

2. CLDS models are easy to fit (§2.3). If Gaussian noise is assumed, then exact latent variable inference (via Kalman smoothing) and fast optimization (via closed-form EM) is possible. Under more realistic noise models (e.g. Poisson), the posterior over latent state trajectories is still log concave and amenable to relatively fast and simple inference routines.

3. CLDS models are expressive. As $\boldsymbol{u}_t$ changes, the parameters of the linear system are allowed to change *non-linearly*. Thus, CLDS can model complex dynamical structures such as ring attractors (§3.2), that are impossible for a vanilla LDS to capture.

4. CLDS models are data efficient. To the extent that LDS parameters change smoothly as a function of $\boldsymbol{u}_t$, we can the recover the parameters of the dynamical system with very few trials per condition (§3.5). In fact, CLDS models can interpolate to make accurate predictions on entirely unseen conditions.

Finally, CLDS models have connections to several existing methods (§4). For example, they can be viewed as a dynamical extension of a Wishart process model [15] and an extension of Gaussian Process Factor Analysis (GPFA; [16]) with a learnable kernel and readout function that can vary across time and conditions. They are also similar to various forms of switching linear dynamical systems models [17, 18, 19]. The key difference is that the "switching" in CLDS models is governed by an observed covariate vector, $\boldsymbol{u}_t$, rather than by a discrete latent process. This makes inference in the CLDS model much more straightforward, albeit at the price of not being fully unsupervised.

## 2   Methods

**Notation**   We use $\mathbf{f}(\cdot) \sim \mathcal{GP}^N(\boldsymbol{m}(\cdot), k(\cdot, \cdot))$ with mean $\boldsymbol{m} : \mathcal{X} \to \mathbb{R}^N$ and kernel $k : \mathcal{X} \times \mathcal{X} \to \mathbb{R}$, to denote samples $\mathbf{f} : \mathcal{X} \to \mathbb{R}^N$ obtained from stacking independent Gaussian processes into an $N$-dimensional vector. Also, $\boldsymbol{I}_D$ is the $D \times D$ identity matrix.

## 2.1 Conditionally Linear Dynamical Systems

Consider an experiment with $N$ neurons recorded over $K$ trials of length $T$. Our dataset consists of the recorded neural firing-rate trajectories $\{\mathbf{y}_{1:T}^k\}_{k=1}^K$, with $\mathbf{y}_t \in \mathbb{R}^N$ at time-step $t \in \{1, \ldots, T\}$, along with the corresponding experimental conditions $\{\boldsymbol{u}_{1:T}^k\}_{k=1}^K$, with $\boldsymbol{u}_t$ in the condition space $\mathcal{U}$. All modeling is done for single trials, and we drop superscripts $k$ where clear henceforth. By experimental conditions, we refer to available neural data covariates, either experimentally set or collected as measurements. These conditions, see Fig. 1a, can vary over time or remain constant (compare §3.4 vs. §3.5). They can also be a step function over time (e.g. animal moving vs. not moving), resulting in a switching-like mechanism between different dynamics.

We model responses $\mathbf{y}_t$ as emissions from a latent time-varying *linear* dynamical system in $\mathbf{x}_t \in \mathbb{R}^D$, with dynamics governed by the conditions $\boldsymbol{u}_t$. Specifically,

$$\mathbf{x}_{t+1} = \mathbf{A}(\boldsymbol{u}_t)\mathbf{x}_t + \mathbf{b}(\boldsymbol{u}_t) + \boldsymbol{\epsilon}_t \tag{1a}$$

$$\mathbf{y}_t = \mathbf{C}(\boldsymbol{u}_t)\mathbf{x}_t + \mathbf{d}(\boldsymbol{u}_t) + \boldsymbol{\omega}_t \tag{1b}$$

over time steps $t \in \{1, \ldots, T\}$ from initial condition $\mathbf{x}_1 \sim \mathcal{N}(\mathbf{x}_1; \mathbf{m}(\boldsymbol{u}_1), Q_1)$, and where $\boldsymbol{\epsilon}_t \sim \mathcal{N}(0, Q)$ and $\boldsymbol{\omega}_t \sim \mathcal{N}(0, R)$ are sources of noise, i.i.d. in time. We assume that the latent variables $\mathbf{x}_t$ follow smooth dynamics defined by time-varying linear matrices $\mathbf{A}(\boldsymbol{u}) \in \mathbb{R}^{D \times D}$ from initial mean $\mathbf{m}(\boldsymbol{u}_1) \in \mathbb{R}^D$, with bias terms $\mathbf{b}(\boldsymbol{u}) \in \mathbb{R}^D$, and emissions governed by $\mathbf{C}(\boldsymbol{u}) \in \mathbb{R}^{N \times D}$ and $\mathbf{d}(\boldsymbol{u}) \in \mathbb{R}^N$, all dependent on $\boldsymbol{u} \in \mathcal{U}$. See graphical depiction in Fig. 1b.

The system in (1a-1b) parameterizes a *family of linear systems* indexed by a continuous variable $\boldsymbol{u}_t$, which importantly is observed. Our approach can be viewed as the linearization in $\mathbf{x}_t$ of a fully nonlinear system in $\mathbf{x}_t$ and $\boldsymbol{u}_t$, under additive noise—we detail this relationship in Appendix §A.2. Most methods treat conditions as additive *inputs*, influencing the dynamics in (1a) via additive terms of the form $\boldsymbol{B}\boldsymbol{u}_t$ for a linear encoding matrix $\boldsymbol{B} \in \mathbb{R}^{D \times |\mathcal{U}|}$. In contrast, the mapping of experimental conditions onto linear dynamics, $\boldsymbol{u} \mapsto \{\mathbf{A}(\boldsymbol{u}), \mathbf{b}(\boldsymbol{u}), \mathbf{C}(\boldsymbol{u}), \mathbf{d}(\boldsymbol{u}), \mathbf{m}(\boldsymbol{u})\}$, is allowed to be nonlinear and learnable. Specifically, we place an approximate Gaussian Process (GP) prior on each entry of the parameters through a finite expansion of basis functions, leveraging regular Fourier feature approximations [20]. For any $\mathbf{M} \in \{\mathbf{A}, \mathbf{b}, \mathbf{C}, \mathbf{d}, \mathbf{m}\}$, we consider a prior

$$\mathbf{M}_{ij}(\boldsymbol{u}) = \sum_{\ell=1}^L \mathrm{w}_\ell^{(ij)} \phi_\ell(\boldsymbol{u}), \quad \mathrm{w}_\ell^{(ij)} \overset{iid}{\sim} \mathcal{N}(0, 1), \tag{1c}$$

over each $i, j$-th entry, truncated at $L \in \mathbb{N}$ basis functions. Each basis function $\phi_\ell : \mathcal{U} \to \mathbb{R}$ is fixed, and the randomness in the prior purely comes from the weights, $\mathrm{w}_\ell^{(ij)}$, which are drawn from a standard normal distribution. When constructed appropriately, the prior in equation (1c) converges to a nonparametric Gaussian process in the limit that $L \to \infty$, with kernel structure determined by the basis functions. Many choices are valid, and we elect to choose basis functions $\{\phi_\ell\}_{\ell=1}^L$ with the goal of expressivity; they are designed to approximate a GP prior of the form $\mathbf{M}_{ij}(\cdot) \sim \mathcal{GP}(0, k_u)$ for the squared exponential kernel $k_u$ with variance $\sigma^2$ and length-scale $\kappa$ [21, eq. 13].

We denote $\mathbf{F} = \{\mathbf{A}, \mathbf{b}, \mathbf{C}, \mathbf{d}, \mathbf{m}\}$ as the set of random functions, and analogously the parameter set $\mathbf{F}(\boldsymbol{u}) = \{\mathbf{A}(\boldsymbol{u}), \mathbf{b}(\boldsymbol{u}), \mathbf{C}(\boldsymbol{u}), \mathbf{d}(\boldsymbol{u}), \mathbf{m}(\boldsymbol{u})\}$ for any condition $\boldsymbol{u} \in \mathcal{U}$. The model distribution

$$p(\mathbf{y}_{1:T}, \mathbf{x}_{1:T} \mid \mathbf{A}, \mathbf{b}, \mathbf{C}, \mathbf{m}, \boldsymbol{u}_{1:T}) = p(\mathbf{y}_{1:T}, \mathbf{x}_{1:T} \mid \mathbf{F}, \boldsymbol{u}_{1:T}) \tag{2}$$

describes a time-varying LDS, conditioned on a parameter sequence set at experimental conditions. Therefore, we refer to the model (1) as a *Conditionally Linear Dynamical System* (CLDS[1]).

## 2.2 CLDS modeling choices

Practitioners can adapt a CLDS model in several ways to suit different applications and modeling assumptions. First, the GP prior can be tuned to trade off model expressivity for interpretability and learnability. In one extreme, as we let $\kappa \to 0$, the LDS parameters change rapidly, nonlinearly as a function of $\boldsymbol{u}$ and become independent per $\boldsymbol{u}$. In the other extreme, if one takes $\kappa \to \infty$, then the LDS parameters become constant (do not change as a function of $\boldsymbol{u}$) and we recover a time-invariant LDS model with autonomous dynamics. In this regime, we could also modify the GP prior over $\mathbf{b}(\cdot)$

---

[1]Our CLDS implementation is available at https://github.com/neurostatslab/clds.

to follow a linear kernel, $k(\boldsymbol{u}, \boldsymbol{u}') = \boldsymbol{u}^\top \boldsymbol{u}'$, resulting in time-invariant LDS with additive dependence on $\boldsymbol{u}_t$. Thus, CLDS models capture classical linear models as a special case. Moreover, the model's prior can be tuned to capture progressively nonlinear dynamics.

A second source of flexibility is the encoding of experimental covariates, $\boldsymbol{u}$. Recall our notation from section 2.1, that $\boldsymbol{u}_t^k$ represents experimental covariates at time $t \in \{1, \dots, T\}$ and trial $k \in \{1, \dots, K\}$. A simple, and broadly applicable, modeling approach would be to set $\boldsymbol{u}_t^k = t$. This achieves a time-varying LDS model in which the GP prior encodes smoothness over time. This is similar in concept to fitting a linear model to data over a sliding time window [22, 23]. However, the CLDS formulation of this idea is fully probabilistic, which has several advantages. For example, one can use a single pass of Kalman smoothing to infer the distribution over the latent state trajectory, $\mathbf{x}_{1:T}^k$, within each trial. It is comparatively non-trivial to average latent state trajectories across multiple LDS models that are independently fit to data in overlapping time windows.

In section 3, we demonstrate more sophisticated examples where $\boldsymbol{u}_t^k$ is specified to track a continuously measured behavioral variable (e.g. head direction or position of an animal) or follow a stepping or ramping function aligned to discrete task events (e.g. a sensory "go cue" or movement onset). Section 4 discusses further connections between CLDS models and existing state space models.

## 2.3 Inference

As mentioned earlier, the conditional distribution in eq. (2) has the advantage of describing a latent LDS, or linear Gaussian state-space model, given estimates of $\mathbf{F}$. As such, we can benefit from analytic tools like Kalman filtering to compute the filtering distributions $p(\mathbf{x}_t \mid \mathbf{y}_{1:t}, \mathbf{F}, \boldsymbol{u}_{1:t})$ and marginal log-likelihood $p(\mathbf{y}_{1:T} \mid \mathbf{F}, \boldsymbol{u}_{1:T})$, and Kalman smoothing for $p(\mathbf{x}_t \mid \mathbf{y}_{1:T}, \mathbf{F}, \boldsymbol{u}_{1:T})$ and the latents posterior mode $\hat{\mathbf{x}}_{1:T}$. We focus on performing maximum-a-posteriori (MAP) inference for these parameters. In principle, it would be a straightforward extension to use variational inference or Markov Chain Monte Carlo to approximate the full posterior over these parameters.

**Conditionally Linear Regression**  As a stepping stone towards our goal of performing MAP inference for $\{\mathbf{A}, \mathbf{b}, \mathbf{C}, \mathbf{d}, \mathbf{m}\}$, consider the model

$$\mathbf{y}_n = \mathbf{M}(\boldsymbol{u}_n)\mathbf{x}_n + \boldsymbol{\epsilon}_n, \qquad \boldsymbol{\epsilon}_n \sim \mathcal{N}(0, \Sigma), \quad \mathbf{M}(\cdot) \sim \mathcal{GP}^{D_1 \times D_2}(0, k_u), \tag{3}$$

given data $\mathbf{y}_n \in \mathbb{R}^{D_1}$, regressors $\mathbf{x}_n \in \mathbb{R}^{D_2}$, conditions $\boldsymbol{u}_n \in \mathcal{U}$, repeats $n \in \{1, \dots, N\}$, noise covariance $\Sigma \succ 0$, and with an approximate GP prior on $\mathbf{M}$ parametrized as in (1c). We refer to the model (3) as *conditionally linear regression*, and our goal is to perform MAP inference for the weights $\{\mathrm{w}_k^{(ij)}\}_{i,j,k}$ for $\mathbf{M}$.

Our approximate basis representation in eq. (1c) implies that each entry is a dot product, $\mathbf{M}_{ij}(\cdot) = \boldsymbol{w}^{(ij)\top}\boldsymbol{\phi}(\cdot)$, where $\boldsymbol{\phi}(\cdot) = (\phi_1(\cdot), \dots, \phi_L(\cdot))^\top \in \mathbb{R}^L$ is our vector of basis-functions evaluations. Therefore, we have

$$\mathbf{M}(\boldsymbol{u})\boldsymbol{X} = \mathbf{W}^\top(\boldsymbol{\phi}(\boldsymbol{u}) \otimes \boldsymbol{X}),$$

for any $\boldsymbol{u} \in \mathcal{U}$ and vector or matrix $\boldsymbol{X} \in \mathbb{R}^{D_2, \cdots}$ of appropriate dimension, with "$\otimes$" the Kronecker product, and with our weights aggregated into the matrix $\mathbf{W}_{D_2\ell+j, i} := \mathrm{w}_\ell^{(ij)}$, $\mathbf{W} \in \mathbb{R}^{D_2 L \times D_1}$. Derivations are provided in Appendix §A.1.1. With this, we can rewrite our regression problem as

$$\mathbf{y}_n = \mathbf{M}(\boldsymbol{u}_n)\mathbf{x}_n + \boldsymbol{\epsilon}_n = \mathbf{W}^\top\mathbf{z}_n + \boldsymbol{\epsilon}_n \tag{4}$$

with $\mathbf{z}_n := \boldsymbol{\phi}(\boldsymbol{u}_n) \otimes \mathbf{x}_n \in \mathbb{R}^{D_2 L}$. Thus, we have reformulated our original model into Bayesian linear regression in an expanded feature space. Namely, the MAP estimate of the weights, $\mathbf{W}_{\mathrm{MAP}}$, is given by

$$\arg\max_{\mathbf{W}} \log p(\mathbf{y}_{1:N} \mid \mathbf{W}, \mathbf{x}_{1:N}, \boldsymbol{u}_{1:N}) + \log p(\mathbf{W}) \tag{5}$$

which is a linear regression problem with regularization from the prior $\log p(\mathbf{W}) = -\frac{1}{2}\|\mathbf{W}\|_F^2$ (up to an additive constant) from (1c). We can analytically solve for the solution (derivations in Appendix §A.1.2), which yields that $\mathbf{W}_{\mathrm{MAP}}$ is the solution to the Sylvester equation

$$\boldsymbol{Z}^\top\boldsymbol{Z}\,\mathbf{W} + \mathbf{W}\Sigma = \boldsymbol{Z}^\top\boldsymbol{Y} \tag{6}$$

with $\boldsymbol{Z} \in \mathbb{R}^{N \times D_2 L}$ our matrix obtained by stacking $\{\mathbf{z}_n\}_{n=1}^N$, and similarly for $\boldsymbol{Y} \in \mathbb{R}^{N \times D_1}$. We see that if $\Sigma = \sigma^2 \boldsymbol{I}_{D_1}$ for some $\sigma > 0$ then we obtain back the familiar looking penalized least squares estimate $\mathbf{W}_{\mathrm{MAP}} = (\boldsymbol{Z}^\top\boldsymbol{Z} + \sigma^2\boldsymbol{I}_{D_1})^{-1}\boldsymbol{Z}^\top\boldsymbol{Y}$.

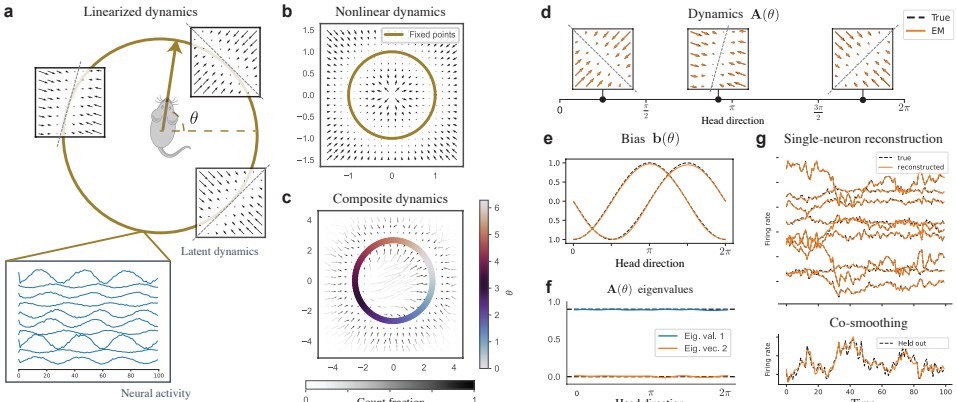

Figure 2: *Head direction synthetic experiment.* (**a**) Schematic of latent dynamics and neural activity about $\theta \in [0, 2\pi)$, the mouse HD, serving as conditions $\boldsymbol{u} = \theta$ in this task. (**b**) True nonlinear flow-field corresponding to the schematic in **a**, computed considering $p(\theta \mid \mathbf{x}) = \delta_{\angle \mathbf{x}}(\theta)$. (**c**) Recovered composite dynamics by CLDS (see text). Grey scale indicates posterior $\hat{\mathbf{x}}_t$ occupancy, and thus our ability to estimate the dynamics. The model fixed points (colored) as a function of $\theta$ form a perfect ring, overlapping with the true fixed points. (**d-e**) Parameter recovery for the dynamics matrix $\mathbf{A}(\theta)$ (**d**) and the bias $\mathbf{b}(\theta)$ (**e**) as functions of head direction $\theta$. (**f**) Recovered eigenvalues of $\mathbf{A}(\theta)$ as a function of $\theta$, true in dashed. (**g**) Co-smoothing reconstruction from the test-set. The firing rate of one neuron is held-out (bottom) while the rest (top) is observed, and we reconstruct accurately the single-neuron firing rates for both the held-in (top) and held-out (bottom) neurons.

**Inference with Expectation Maximization**  We can leverage the above to perform MAP inference for $\mathbf{F} = \{\mathbf{A}, \mathbf{b}, \mathbf{C}, \mathbf{d}, \mathbf{m}\}$ with the *Expectation-Maximization* (EM) algorithm [24, 4]. In the *E*-step we obtain estimates of the moments of the latents with Kalman-smoothing, which then place us in a setting akin to eq. (3) with sufficient statistics as data and regressors. We can then perform closed-form M-steps with our updates in eq. (6). We provide in Appendix §A.1.3 an example of the associated derivations with these E- and M-steps for the joint update for $\mathbf{A}(\cdot)$ and $\mathbf{b}(\cdot)$. The resulting EM algorithm has several advantages: (1) all E- and M-steps are analytic, (2) the E-step provides us with exact (penalized) marginal log-likelihood calculations, and (3) the algorithm gives monotonic gradient ascent guarantees of the marginal log-likelihood (resp. log posterior) objective.

We initialize the EM algorithm at samples from our GP priors for $\mathbf{F}$. With the EM algorithm we also learn the covariance parameters $\{Q_1, Q, R\}$. The hyper-parameters $\{L, \kappa, \sigma\}$ for the GP priors and the dimensionality of the latents $D$ are determined through performance on held-out test sets from 80/20 trial splits on all experiments unless specified.

**Extensions to non-Gaussian likelihoods**  The closed form EM updates are only applicable when the observation distribution of $\mathbf{y}_t$ conditioned on $\mathbf{x}_t$ and $\boldsymbol{u}_t$ (1b) is Gaussian. This assumption is common to existing methods (e.g. [16, 25]), though to model spike count data directly, most work utilize Poisson (e.g. [26]) and COM-Poisson [27] likelihoods. Such emission likelihoods of interest have log concave distributions, for which inference in CLDS models would remain tractable. Indeed, conditioned on $\boldsymbol{u}_{1:T}$, the log posterior density associated with $\mathbf{x}_{1:T}$ is equal to a sum of concave terms up to an additive constant [1]. Thus, we can use standard optimization routines to identify a MAP estimate of $\mathbf{x}_{1:T}$ with theoretical guarantees, and leverage it for approximate EM [26].

## 3 Experiments

### 3.1 Setup

**Metrics**  For a given trajectory $\{\mathbf{y}_{1:T}, \boldsymbol{u}_{1:T}\}$, we denote as *data reconstruction* the mean emission $\mathbb{E}\left[\mathbf{y}_{1:T} \mid \hat{\mathbf{x}}_{1:T}, \mathbf{F}(\boldsymbol{u}_{1:T})\right] = (\mathbf{C}(\boldsymbol{u}_1)\hat{\mathbf{x}}_1, \ldots, \mathbf{C}(\boldsymbol{u}_T)\hat{\mathbf{x}}_T)$ from a the posterior mode $\hat{\mathbf{x}}_{1:T}$, computed with Kalman smoothing, given the observations $\mathbf{y}_{1:T}$ and parameters $\mathbf{F}(\boldsymbol{u}_{1:T})$. As our primary metric, we use *co-smoothing* [28] to evaluate the ability of models to predict held-out single-neuron

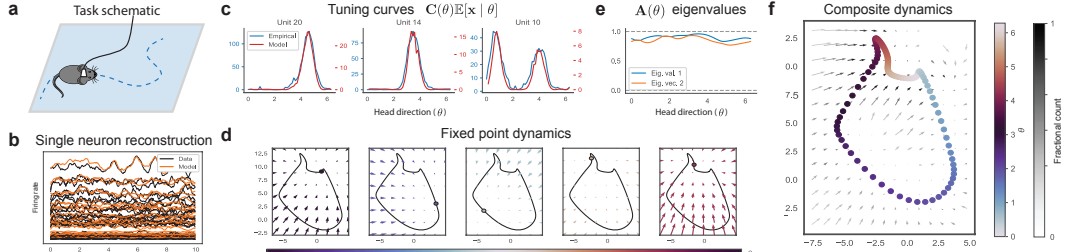

Figure 3: (**a**) Schematic of mouse foraging in an open environment. We have access to $\theta_t \in [0, 2\pi)$ the mouse HD in time $t$, which we use as conditions $\boldsymbol{u}_t$ just like Fig. 2. (**b**) Model reconstruction on the whole dataset recovers the true data. We plot single-neuron traces, averaged over 10s trials. (**c**) Model tuning curves over head direction $\theta$, obtained as $\mathbf{C}(\theta)\mathbb{E}[\mathbf{x} \mid \theta]$, recover the empirical tuning curves. Plotted for the top three units in firing rate norm. (**d**) Dynamics around each fixed point in **x**-state space as a function of head direction $\theta$, with the solid-line representing the complete set of fixed points. (**e**) Eigenvalues and angles of eigenvectors of $\mathbf{A}(\theta)$ as a function of $\theta$. (**f**) Composite dynamics in $\mathbf{x}_t$-space, with overlaid colored model fixed points as a function of $\theta$.

activity. Specifically, for the top 5 neurons with highest variance from the test set, we compute the coefficient of determination $R^2$ between the true and reconstructed single-neuron firing rate, obtained by performing data reconstruction using only the other neurons.

**Composite dynamics**    The latent dynamical system (1a) depends on the condition $\boldsymbol{u}_t$, which can make visualizations challenging. Building on the idea that CLDS models decompose a nonlinear dynamical system into linearizations governed by $\boldsymbol{u}$ (see App. §A.2), we aim to approximate the general nonlinear system by marginalizing out $\boldsymbol{u}_t$, conditioned on $\mathbf{x}_t$. That is,

$$\mathbf{x}_{t+1} = \boldsymbol{g}(\mathbf{x}_t) + \epsilon_t := \mathbb{E}_{p(\boldsymbol{u}|\mathbf{x}_t)}\left[\mathbf{A}(\boldsymbol{u})\mathbf{x}_t + \mathbf{b}(\boldsymbol{u})\right] + \epsilon_t, \tag{7}$$

which we define as the *composite dynamical system*. Intuitively, we expect this to provide a good approximation to the underlying nonlinear dynamics when $\boldsymbol{u}_t$ and $\mathbf{x}_t$ tightly co-determine each other—i.e., when the encoding $p(\mathbf{x}_t \mid \boldsymbol{u}_t)$ and decoding $p(\boldsymbol{u}_t \mid \mathbf{x}_t)$ conditional distributions have low variance (see App. §A.2.2). In practice, we estimate the expectation in (7) by averaging over the $\boldsymbol{u}_n$ associated with binned values $\hat{\mathbf{x}}_n$ of the posterior mode given a trajectory $\{\mathbf{y}_{1:T}, \boldsymbol{u}_{1:T}\}$. Thus our ability to accurately estimate the flow-field around a given point under this method depends on how many posterior samples pass by it, which we report alongside the composite dynamics plots.

**Model baselines**    For model comparison, we use as baselines (1) a standard **LDS** model and (2) a more flexible **gpSLDS** [19] (implemented in a way to include linear boundaries to encompass the rSLDS [29]), both with additive inputs of the form $\boldsymbol{Bu}_t$ in the latent dynamics, and (3) **LFADS** [9] model with controller inferred-inputs as a fully nonlinear alternative. For all, we consider Gaussian observation model to fit directly to the firing rates. See Appendix §B.1 for implementation details.

### 3.2    Synthetic head-direction ring attractor

We start by considering a synthetic experiment of head direction neural dynamics. We conceptualize latent dynamics that capture the head direction (HD) of the animal, with attractor dynamics about a HD-dependent fixed point—see schematic in Fig. 2**a**. This synthetic experiment is designed to represent a nonlinear system decomposed as linear systems, per HD serving as the condition. We plot in Fig. 2**b** what the resulting, "composite dynamics" (see §3.1), nonlinear flow-field would be, assuming the latent state encodes the head direction exactly. The generative dynamics are an instance of a CLDS model by construction, so this example allows us to explore recovery performance.

Concretely, let $\theta_t \in [0, 2\pi)$ denote heading direction at time step $t$, which we treat as our conditions $\boldsymbol{u}_t := \theta_t$. To build a ring attractor, we parametrize two orthogonal unit vectors

$$\boldsymbol{e}_1(\theta) = [\cos(\theta) \quad \sin(\theta)], \quad \boldsymbol{e}_2(\theta) = [-\sin(\theta) \quad \cos(\theta)], \tag{8}$$

that describe the position on the ring and the tangent vector respectively. We design (i.e. impose) that the system converges to a stable fixed point at $\boldsymbol{e}_1(\theta)$, and at head direction $\theta$ we approximately

integrate speed input along the subspace spanned by $e_2(\theta)$. To do this, we define $\boldsymbol{A}(\theta)$ to be a rank-one matrix that defines a leaky line attractor, with attracting (i.e. contracting) dynamics along the orthogonal $e_1(\theta)$. For a hyperparameter $0 < \epsilon < 1$ define:

$$\boldsymbol{A}(\theta) := (1 - \epsilon)\boldsymbol{e}_2(\theta)\boldsymbol{e}_2(\theta)^\top, \quad \boldsymbol{b}(\theta) := \boldsymbol{e}_1(\theta). \tag{9}$$

Completing the model description, we assume that the firing rate of individual neurons is given by a linear readout as $\mathbf{y}_{t,i} = \boldsymbol{C}_{i,:}(\theta_t)^\top \mathbf{x}_t + \boldsymbol{\omega}_t$ for neuron $i$ at time $t$, with $\mathbf{d}(\boldsymbol{u}_t) = \mathbf{0}$ and $\boldsymbol{C}_{i,:}$ is sinusoidal bump tuning curve function (see App. §B.2). Finally, we sampled trials of length $T = 100$ with $N = 10$ neurons, generating the evolution of the heading direction as a random walk, $\theta_t \sim \mathcal{N}(\theta_{t-1}, 0.5^2)$, and initialize at the origin $\mathbf{x}_1 \sim \mathcal{N}(0, 1)$.

We report our recovery results in Fig. 2, fixing the decoding matrix $\mathbf{C}(\cdot)$ to a known value as to avoid non-identifiability considerations. We refer to App. §B.2 for recovery plots of $\mathbf{C}$ when fitted. First, we observed that we can generally recover the nonlinear flow-field, plotting in Fig. 2**c** the composite dynamics obtained from the posterior trajectories. Second, for a more parameter-based account of the recovery, we plot in Fig. 2**d-e** the varying biases $\mathbf{b}(\theta)$ and dynamics matrices $\mathbf{A}(\theta)$ as functions of the head direction $\theta$—we recovered with high-fidelity the true parameters. This recovery translated into the properties of the dynamics such as the recovered eigenvalues of $\mathbf{A}(\theta)$ in Fig. 2**f**. Third, we observed that the test data single-neuron reconstruction (Fig. 2**g**) recovers the true observations, and the model was able to accurately ($R^2 = 0.86$) reconstruct a held out neuron from this test-set through co-smoothing. Finally, we report in Appendix §B.2 a study on the impact of observational noise on our quality of fit—we find reliable recovery even as the signal-to-noise ratio degrades.

### 3.3 Neural circuit model of ring-attractor dynamics

We provide an alternate synthetic ring attractor experiment based on the neuroscience literature [30]. It is meant to test the CLDS and its inference in a synthetic data modality with known underlying computation but which, importantly, is not generated from a CLDS model.

We write a model of continuous ring attractor dynamics with bumps of activity integrating angular velocity [31]. In this model, high-dimensional and non-normal attractor dynamics are implemented through HD-preferential units arranged in a "ring", with short-range excitation and long-range inhibition forming a bump of activity for a specific unit in a specific HD, and with a HD velocity $v(t) = \dot{\theta}(t)$ integrator causing a shift in the location of the bump. Let $\mathbf{y}(\phi, t)$ denote the network activity of cells with preferred head-direction $\phi \in [0, 2\pi)$ at time $t$. The dynamics follow the stochastic partial differential equation

$$\frac{\partial}{\partial t}\mathbf{y} = \tau \left[ -\mathbf{y} + f(w * \mathbf{y}) + v\,\frac{\partial}{\partial \phi}\mathbf{y} \right] + \sigma\xi \tag{10}$$

for $\xi(\phi, t)$ a white noise process on the ring. Here, $f$ is a nonlinearity, $w$ a "Mexican-hat" convolution filter, and $\tau > 0$ a time constant. For implementation purposes, $\mathbf{y}$ is discretized to $\mathbf{y}_t \in \mathbb{R}^N$, $N = 32$, for $t \in \{1, \ldots, T\}$, with each unit $[\mathbf{y}]_i$ having preferred directions $\phi_i$ regularly spanning the interval $[0, 2\pi)$. See model and implementation details in Appendix §B.3. We use $\mathbf{y}$ as our observations and head-direction $\theta$ as conditions again, generated the same way as in our previous synthetic HD model.

The underlying model captures high-dimensional ring attractor dynamics. When fitting a CLDS model, we expect the low-dimensional latent dynamics to capture the ring-attractor structure and the nonlinearity of the dynamics, with a linear emission model bringing it back to the high-dimensional activity. This is precisely what we found when fitting the CLDS—see Appendix §B.3. The results make us confident that the CLDS can capture this nonlinear structure even under model mismatch.

### 3.4 Mouse head-direction dynamics

Next, we turned to the analysis of antero-dorsal thalamic nucleus (ADn) recordings from Ref. [32] of the mouse HD system in mice foraging in an open environment (Fig. 3**a**). We considered neural activity from the "wake" period, binned in 50ms time-bins, then processed to firing rates and separated into 10s trials. As with the synthetic experiment of the previous section, we treat the recorded head-direction $\theta_t \in [0, 2\pi)$ as conditions $\boldsymbol{u}_t = \theta_t$.

We recovered single-neuron firing rates with high accuracy (Fig. 3**b**) through data reconstruction. We further validated our fit by computing the empirical tuning curves, which our model recovered almost

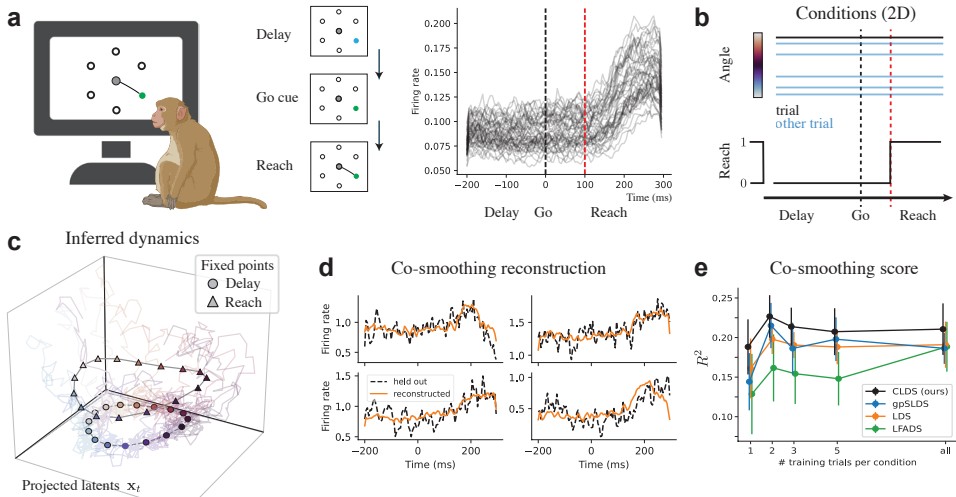

Figure 4: *Macaque reaching experiment*. (**a**) Task schematic (left) and population-averaged firing rates per trial (right). (**b**) 2D conditions, with trial orientation $\theta$, and reach variable $z_t \in \{0, 1\}$ switching at ramp onset. (**c**) 3D projection of the 5 dimensional latents used, projected as to align best with condition decoding. We show the model fixed-points per reach angle $\theta$ and reach condition $z$, plotted over posterior mean trajectories per trial. (**d**) Co-smoothing reconstruction of single held-out neurons from the test-set. (**e**) Co-smoothing $R^2$ per model as a function of the number of trials used per reach angle during training. Error bars indicated std. around the mean over 5 initialization seeds.

exactly (Fig. 3**c**). The model tuning curves are given by

$$\mathbb{E}\left[\mathbf{y}_i \mid \theta\right] = \mathbb{E}\left[\mathbb{E}\left[\mathbf{y}_i \mid \mathbf{x}, \theta\right]\right] = \boldsymbol{C}_{i,:}(\theta)^\top \mathbb{E}\left[\mathbf{x} \mid \theta\right], \tag{11}$$

which follows from the law of total expectation. The later quantity $\mathbb{E}[\mathbf{x} \mid \theta]$ represents the expected encoding of the conditions $\theta$, which we estimate by averaging posterior trajectories, obtained with Kalman smoothing, over (binned) $\theta$ given corresponding firing rates.

Finally, we analyzed the learned latent dynamics. Like the synthetic example, we identified a ring attractor structure (Fig. 3**d**). Unlike the synthetic example, per the eigenvalues of $\mathbf{A}(\theta)$ in Fig. 3**e**, this ring attractor is composed of HD-dependent fixed points as opposed to line attractors.

### 3.5 Macaque center-out reaching task

Finally, we analyzed neural recordings of dorsal premotor cortex (PMd) in macaques performing center-out reaching task (Fig. 4**a**) from Ref. [33]. In contrast to the previous experimental conditions, we consider here two-dimensional conditions $\boldsymbol{u}_t^k = (\theta^k, z_t)$, where $\theta^k \in [0, 2\pi]$ is the instructed reach angle, constant per trial $k$, and $z_t \in \{0, 1\}$ indicates the task reach condition (see Fig. 4**b**) set at 0 during the delay and 1 at 100ms past the go-cue, at the onset of the movement-related firing rate ramp Fig. 4**a**-(right). Discrete-valued conditions, such as the reach onset $z_t \in \{0, 1\}$, are considered as supported on a continuous interval. The correlation between such discrete points is determined by the length-scale parameter $\kappa$, which we've set to $\kappa = 0.5$ from a hyperparameter search. More details on hyperparameters and data-preprocessing can be found in Appendix §B. Finally, we use a fixed emission matrix $\boldsymbol{C}$ and let the latent dynamics capture the dependency on experimental conditions through $\mathbf{A}(\boldsymbol{u})$ and $\mathbf{b}(\boldsymbol{u})$.

We found the latent dynamics to encode the conditions through attracting fixed-points during both the delay and reach periods. We show in Fig. 4**c** the projection of the $D = 5$ latent dimensions along the 3-dimensional subspace most aligned (i.e. best decoding) with the experimental conditions, following common analyses [33]—we observed clear aligned rings of fixed points from delay to reach. In CLDS models, we obtain the fixed points by simply solving for $\boldsymbol{x}^*(\boldsymbol{u})$ satisfying $(\boldsymbol{I} - \mathbf{A}(\boldsymbol{u}))\boldsymbol{x}^* = \mathbf{b}(\boldsymbol{u})$ for any $\boldsymbol{u}$, in contrast to numerical fixed-point methods usually employed [34].

We performed co-smoothing (see §3.1) to evaluate the model. We recovered with good accuracy single held-out neurons from the validation set excluded from training (Fig. 4**d**). We then compared

the performance of the CLDS against the baseline models, exploring further how each fares in low-data regimes. We report in Fig. 4**e** the co-smoothing $R^2$ per model, computed as a function of the number of trials used in each reach-angle $\theta^k$, averaged over 5 random seeds. We found that the CLDS outperformed other models, displaying the highest difference at 1 training trial per condition. Introducing more training trials per condition, the gpSLDS model sometimes approached the performance of the CLDS. While the LFADS model showed progressively better performance that did not plateau yet, it nonetheless underperformed in these low data regimes.

## 4   Related work

**Wishart process models**   CLDS models capture the dependence of neural responses $\mathbf{y}_{1:T}$ on continuous experimental conditions $\boldsymbol{u}_{1:T}$. Ref. [15] investigated a very similar problem, focusing on single-trial responses $\mathbf{y}_k$ to continuous experimental conditions $\boldsymbol{u}_k \in \mathcal{U}$. They use a conditional Gaussian model for responses $\mathbf{y}_k$ given conditions $\boldsymbol{u}_k$

$$\mathbf{y}_k \mid \boldsymbol{u}_k \sim \mathcal{N}(\mathbf{y}; \mu(\boldsymbol{u}_k), \Sigma(\boldsymbol{u}_k)) \tag{12a}$$

and they place Gaussian process and Wishart process [35] priors on the mean and covariance functions

$$\mu(\cdot) \sim \mathcal{GP}^M(0, k_\mu), \qquad \Sigma(\boldsymbol{u}) = \mathbf{U}(\boldsymbol{u})\mathbf{U}(\boldsymbol{u})^\top + \Lambda(\boldsymbol{u}) \tag{12b}$$

with $\mathbf{U}(\cdot) \sim \mathcal{GP}^{M \times p}(0, k_\Sigma)$ and $\Lambda(\cdot) \sim \mathcal{GP}^M(0, k_\Lambda)$ for chosen kernel functions $\{k_\mu, k_\Sigma, k_\Lambda\}$. The hyper-parameter $p \in \mathbb{N}$ determines the low-rank structure of $\Sigma$.

For a single time step $t = T = 1$ and assuming a degenerate prior $\mathbf{m}(\boldsymbol{u}_1) = \mathbf{0}$, the marginal distribution of $\mathbf{y}_1$ conditioned on $\boldsymbol{u}_1$ in our system (1) reads

$$\mathcal{N}\left(\mathbf{d}(\boldsymbol{u}_1), \mathbf{C}(\boldsymbol{u}_1)Q_1\mathbf{C}(\boldsymbol{u}_1)^\top + R\right). \tag{13}$$

which can be compared with (12). The models are analogous with $\mu(\boldsymbol{u}) = \mathbf{d}(\boldsymbol{u})$ and with the CLDS emission matrix $\mathbf{C}(\boldsymbol{u})$ serving as the Wishart process prior decomposition matrix $\mathbf{U}(\boldsymbol{u})$, right-scaled by $Q_1^{1/2} \in \mathbb{R}^{D \times D}$. This makes the parameter $p = D$ now bear meaning as the dimensionality of the latents $\mathbf{x} \in \mathbb{R}^D$, akin to factor analysis. Thus, we can view CLDS models as a direct extension of Wishart process models that capture condition-dependent dynamics across multiple time steps.

**Markovian GPs**   Latent GP models [36, 37], such as GPFA [16], are widely used in neuroscience. One can view MAP inference in a CLDS model as optimizing a kernel that defines a latent GP prior. Taking this a step further, the stationary dynamics of an AR-1 process (i.e. linear dynamical system) can be expressed as draws from a GP [16, 38]. Vice versa, all stationary, real-valued, and finitely differentiable GPs admit a representation in terms of linear state space models [39, 40], a relationship that has proven useful for both modeling and aiding GP inference [40, 41, 42]. We expand upon this direction by allowing the dynamics to vary not only across time but over conditions too. For a fixed set of LDS parameters at experimental covariates, $\boldsymbol{F}(\boldsymbol{u}_{1:T})$, the distribution of latent states in a CLDS are jointly Gaussian. Thus, a set of LDS parameters induces a (generally non-stationary) GP prior on the latent trajectories. In this view, the GP prior we place over the parameters of the LDS can be seen as a hyperprior over the latent dynamical process.

**Switching dynamical systems**   A second class of relevant models generalizing the LDS are *Switching LDS* models (SLDS; [43, 44, 17]). SLDS models consist of a discrete latent state $z_t$, with finite Markov chain dynamics, dictating the dynamics matrix $\boldsymbol{A}^{(z_t)}$. This switching behavior can be mimicked in our setting if the condition space is discrete (see, e.g., §3.5). We can take the relationship a step further by embedding the discrete process in the continuous dynamics parameter space of $\boldsymbol{A}^{(z_t)} \in \mathbb{R}^{D \times D}$. In a similar line of thinking as with Markovian GPs, we show in Appendix §A.3 how a one-dimensional SLDS model of dynamics

$$p(z_{t+1} = i \mid z_t = j) = P_{ij}, \quad x_{t+1} = a^{(z_t)}x_t + \epsilon_t,$$

is equivalent, up to the first two moments of the stochastic process $a := a^z$, to a CLDS model with

$$a(\cdot) \sim \mathcal{GP}\left(\pi^\top \boldsymbol{a}, \boldsymbol{a}^\top \left(P^{|t_j - t_i|}\mathrm{diag}(\pi) - \pi\pi^\top\right)\boldsymbol{a}\right),$$

over time conditions $\boldsymbol{u}_t = t$, for $\boldsymbol{a}$ the vector of values taken by $a^{(z_i)}$ and $\pi$ the stationary distribution of the $z_t$ discrete state process. In a similar vein, previous work has considered the discrete states

$z_t$ dictating the dynamics to live on a continuous support [45], however without leveraging this continuity in the parameters $\mathbf{A}(\cdot)$ themselves.

The *recurrent SLDS* (rSLDS) model [29, 46, 47] takes an important departure from the SLDS by leveraging the continuous latent states $\mathbf{x}_t$ to guide the discrete state transitions. Recent work [48] use this dependency but turn to the linearization of nonlinear systems with $\mathbf{x}$-space fixed points as guide for the linear dynamics. In contrast, we linearize based on observed conditions (see App. A.2).

**Smoothly varying dynamical systems models**   Switching LDS models can be contrasted with models that smoothly interpolate between dynamical parameter regimes. The simplest example of this would be time-varying linear models (e.g. [22]); CLDS models are a generalization of this idea that comes with several advantages (see §2.2). Recent work [49] relaxed switching dynamics to be subject to an approximately continuous and latent time warping factor. More similar to CLDS models are smoothly varying latent linear dynamical models, such as the gpSLDS [19], which relax the discrete switching in rSLDS models to allow smoothly varying soft mixtures of linear dynamics, and the dLDS [50], which learns a dictionary of linear dynamics combined with time-varying coefficients. Again, the primary difference is that CLDS models achieve a similar effect but utilize observed experimental covariates to infer these dynamical transitions. This approach is closer to previous work [51] using neural network architectures to parametrize input-dependent linear recurrent dynamics.

## 5   Conclusion

In this work, we extend classical linear-Gaussian state space models of neural dynamics. Our results suggest that these models are competitive with modern methods when the dynamical parameters vary smoothly as a function of available covariates. Like classical linear models, CLDS models are easy to fit and interpret. Our main technical contribution was to introduce an approximate GP prior over system parameters and show that this leads to closed-form inference under a Gaussian noise model.

While these results are promising, CLDS models have limitations that merit consideration. To start, we did not extend to non-Gaussian observations to model spike counts directly, nor leveraged the prior structure on the coefficients $\mathbf{F}$ to convey more complete parameter posteriors—we discuss both of these options in the text, and see them as essential avenues for future work. Furthermore, as their name implies, CLDS models assume conditionally linear latent dynamics. We detail in Appendix §A.2 how our approach can be viewed as a first-order (in $\mathbf{x}$) approximation to a full nonlinear system of the form

$$\mathbb{E}[\mathbf{x}_{t+1} \mid \mathbf{x}_t, \boldsymbol{u}_t] = \boldsymbol{f}_x(\mathbf{x}_t, \boldsymbol{u}_t), \quad \mathbb{E}[\mathbf{y}_t \mid \mathbf{x}_t, \boldsymbol{u}_t] = \boldsymbol{f}_y(\mathbf{x}_t, \boldsymbol{u}_t).$$

We derive upper bounds for the approximation error between our linearized dynamics and this nonlinear system, focusing of $\boldsymbol{f}_x$ and showing that the quality of this approximation is guaranteed to be small if $\boldsymbol{f}_x$ is well-behaved and the conditional covariance in $\mathbf{x}_t$ given $\boldsymbol{u}_t$ is small. These same guarantees help shed light onto limitations. First, these models rely on observing a time series of experimental covariates $\boldsymbol{u}_{1:T}$ and performance would suffer if the covariates are corrupted, such as during forecasting or with partial observations. Second, the model assumes linear dynamics conditioned on $\boldsymbol{u}_t$. We believe this is a good approximation in many settings of interest, particularly when there is strong tuning in $\mathbf{x}$ to sensory or behavioral variables $\mathbf{u}$, and expect CLDS models to struggle when they are loosely correlated (e.g. cognitive tasks with long periods of internal deliberation). In these scenarios, we expect that more nonlinear approaches will outperform CLDS models when given access to large amounts of data. Nevertheless, neural recordings are often trial-limited in practice [52]. We therefore view CLDS models as a broadly applicable modeling tool for many neuroscience applications.

Future work could extend CLDS models to overcome these limitations, such as handling partially observed covariates, $\boldsymbol{u}_t^k$. Since CLDS models can be viewed as a dynamical extension of Wishart process models (see §4), future work could also apply this method to infer across-time noise correlations [reviewed in 53], in addition to classical across-trial noise correlations. Recent work [54] shows how across-time correlations can be used to quantify similarity in dynamical systems—a topic that has recently attracted strong interest [55]. CLDS models are a potentially attractive framework for tackling the unsolved challenge of estimating this high-dimensional correlation structure in trial-limited regimes.

## Acknowledgments

VG was supported by the Porter Ogden Jacobus Fellowship at Princeton University and by doctoral scholarships from the Natural Sciences and Engineering Research Council of Canada (NSERC) and the Fonds de recherche du Québec – Nature et technologies (FRQNT). JWP was supported by grants from the Simons Collaboration on the Global Brain (SCGB AWD543027), the NIH BRAIN initiative (9R01DA056404-04), an NIH R01 (NIH 1R01EY033064), and a U19 NIH-NINDS BRAIN Initiative Award (U19NS104648). AHW was supported by the NIH BRAIN initiative (1RF1MH133778).

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

# A Modeling

## A.1 Inference

### A.1.1 Basis space form

We want to show

$$\mathbf{M}(\boldsymbol{u})\boldsymbol{X} = \mathbf{W}^{\top}(\boldsymbol{\phi}(\boldsymbol{u}) \otimes \boldsymbol{X})$$

for $\mathbf{M}$ defined as in (1c) with basis functions $\{\phi_\ell\}_{\ell=1}^{L}$, $\boldsymbol{u} \in \mathcal{U}$, and $\boldsymbol{X} \in \mathbb{R}^{D_2 \times D_3}$, $D_3 \in \mathbb{N}$. For each $i, j$-th entry,

$$
\begin{aligned}
[\mathbf{M}(\boldsymbol{u})\boldsymbol{X}]_{ij} &= \sum_{k=1}^{D_2} [\mathbf{M}(\boldsymbol{u})]_{ik} [\boldsymbol{X}]_{kj} \\
&= \sum_{k=1}^{D_2} \sum_{\ell=1}^{L} \mathrm{w}_\ell^{(ik)} \phi_\ell(\boldsymbol{u}) \boldsymbol{X}_{kj} \\
&= \sum_{k=1}^{D_2} \sum_{\ell=1}^{L} \mathrm{w}_\ell^{(ik)} (\boldsymbol{\phi}(\boldsymbol{u}) \otimes \boldsymbol{X})_{D_2\ell+k,j} \\
&= \sum_{k=1}^{D_2} \sum_{\ell=1}^{L} \mathbf{W}_{i,D_2\ell+k}^{\top} (\boldsymbol{\phi}(\boldsymbol{u}) \otimes \boldsymbol{X})_{D_2\ell+k,j} \\
&= \mathbf{W}_{i,:}^{\top} (\boldsymbol{\phi}(\boldsymbol{u}) \otimes \boldsymbol{X})_{:,j} \\
&= \left[\mathbf{W}^{\top} (\boldsymbol{\phi}(\boldsymbol{u}) \otimes \boldsymbol{X})\right]_{ij}
\end{aligned}
$$

as desired, where we've defined $\mathbf{W}_{D_2\ell+k,i} := \mathrm{w}_\ell^{(ik)}$.

### A.1.2 Least squares derivation for Conditionally Linear Regression

Recall

$$\mathbf{M}(\boldsymbol{u})\boldsymbol{X} = \mathbf{W}^{\top}(\boldsymbol{\phi}(\boldsymbol{u}) \otimes \boldsymbol{X}), \qquad \mathbf{W} \in \mathbb{R}^{D_2 L \times D_1}$$

for $\mathbf{M}(\boldsymbol{u}) \in \mathbb{R}^{D_1 \times D_2}$, $\boldsymbol{X} \in \mathbb{R}^{D_2 \times M}$. In particular, $\mathbf{M}(\boldsymbol{u}_n)\mathbf{x}_n = \mathbf{W}^{\top}\mathbf{z}_n$. What follows are standard least-squares derivations for matrix coefficients with matrix regularization, which we include for completeness.

Our posterior objective reads as

$$
\begin{aligned}
\log p(\mathbf{W} \mid \mathbf{y}_{1:N}, \mathbf{x}_{1:N}, \boldsymbol{u}_{1:N}) &\propto \log p(\mathbf{y}_{1:N} \mid \mathbf{W}, \mathbf{x}_{1:N}, \boldsymbol{u}_{1:N}) + \log p(\mathbf{W}) \\
&= \sum_{n=1}^{N} \log p(\mathbf{y}_n \mid \mathbf{W}, \mathbf{x}_n, \boldsymbol{u}_n) + \log p(\mathbf{W}).
\end{aligned}
$$

We have

$$
\begin{aligned}
\log p(\mathbf{y}_n \mid \mathbf{x}_n, \boldsymbol{u}_t) &= -\tfrac{1}{2}(\mathbf{y}_n - \mathbf{W}^{\top}\mathbf{z}_n)^{\top}\Sigma^{-1}(\mathbf{y}_n - \mathbf{W}^{\top}\mathbf{z}_n) - c \\
&= -\tfrac{1}{2}\mathrm{Tr}\left[(\mathbf{y}_n - \mathbf{W}^{\top}\mathbf{z}_n)^{\top}\Sigma^{-1}(\mathbf{y}_n - \mathbf{W}^{\top}\mathbf{z}_n)\right] - c \\
&= -\tfrac{1}{2}\mathrm{Tr}\left[\Sigma^{-1}(\mathbf{y}_n - \mathbf{W}^{\top}\mathbf{z}_n)(\mathbf{y}_n - \mathbf{W}^{\top}\mathbf{z}_n)^{\top}\right] - c \\
&= -\tfrac{1}{2}\left(\mathrm{Tr}\left[\Sigma^{-1}\mathbf{y}_n\mathbf{y}_n^{\top}\right] - 2\mathrm{Tr}\left[\Sigma^{-1}\mathbf{W}^{\top}\mathbf{z}_n\mathbf{y}_n^{\top}\right] + \mathrm{Tr}\left[\Sigma^{-1}\mathbf{W}^{\top}\mathbf{z}_n\mathbf{z}_n^{\top}\mathbf{W}\right]\right) - c.
\end{aligned}
$$

with the normalizing constant $c = \tfrac{1}{2}\log|2\pi\Sigma|$.

To optimize this expression with respect to $\mathbf{W}$, we consider the zeros of the derivative

$$
\begin{aligned}
\frac{\partial}{\partial\mathbf{W}}\log p(\mathbf{y}_n \mid \mathbf{x}_n, \boldsymbol{u}_n) &= \frac{\partial}{\partial\mathbf{W}}\mathrm{Tr}\left[\mathbf{W}^{\top}\mathbf{z}_n\mathbf{y}_n^{\top}\Sigma^{-1}\right] - \frac{1}{2}\frac{\partial}{\partial\mathbf{W}}\mathrm{Tr}\left[\Sigma^{-1}\mathbf{W}^{\top}\mathbf{z}_n\mathbf{z}_n^{\top}\mathbf{W}\right] \\
&= \mathbf{z}_n\mathbf{y}_n^{\top}\Sigma^{-1} - \mathbf{z}_n\mathbf{z}_n^{\top}\mathbf{W}\Sigma^{-1},
\end{aligned}
$$

and

$$\log p(\mathbf{W}) = -\tfrac{1}{2}\|\mathbf{W}\|_F^2 \quad \implies \quad \tfrac{\partial}{\partial \mathbf{W}}\log p(\mathbf{W}) = -\mathbf{W}.$$

Taken together, we thus have that the stationary point of the posterior satisfies

$$\sum_{n=1}^{N}\left(\mathbf{z}_n \mathbf{y}_n^\top \Sigma^{-1} - \mathbf{z}_n \mathbf{z}_n^\top \mathbf{W}\Sigma^{-1}\right) - \mathbf{W} = 0.$$

Define $Y \in \mathbb{R}^{N\times D_1}$, $Z \in \mathbb{R}^{N\times D_2 L}$ by row-wise stacking $\mathbf{y}_n$ and $\mathbf{z}_n$ respectively, and note that $\sum_n \mathbf{z}_n \mathbf{y}_n^\top = Z^\top Y$. We get

$$Z^\top Y \Sigma^{-1} - Z^\top Z \mathbf{W}\Sigma^{-1} - \mathbf{W} = 0$$
$$\implies \quad Z^\top Z \mathbf{W} + \mathbf{W}\Sigma = Z^\top Y$$

as desired.

### A.1.3 Joint dynamics and bias EM update in weight-space

Here we detail how one EM update for the parameters governing $\{\mathbf{A}, \mathbf{b}\}$ is carried out. Using the function-space weights $\mathbf{W}_A \in \mathbb{R}^{DL\times D}$ and $\mathbf{W}_b \in \mathbb{R}^{L\times D}$, the dynamics read

$$\mathbf{x}_{t+1} = \mathbf{A}(\boldsymbol{u}_t)\mathbf{x}_t + \mathbf{b}(\boldsymbol{u}_t) + \epsilon_t$$
$$= \mathbf{W}_A^\top \underbrace{(\boldsymbol{\phi}(\boldsymbol{u}_t) \otimes \mathbf{x}_t)}_{\mathbf{z}_t} + \mathbf{W}_b^\top \boldsymbol{\phi}(\boldsymbol{u}_t) + \epsilon_t$$

Define $\mathbf{z}_t = \boldsymbol{\phi}(\boldsymbol{u}_t) \otimes \mathbf{x}_t \in \mathbb{R}^{LD}$, and note

$$\mathbb{E}_{\mathbf{x}_t}[\mathbf{z}_t] = \boldsymbol{\phi}(\boldsymbol{u}_t) \otimes \mathbb{E}_{\mathbf{x}_t}[x_t]$$
$$\mathbb{E}_{\mathbf{x}_t, \mathbf{x}_{t+1}}[\mathbf{z}_t \mathbf{x}_{t+1}^\top] = \boldsymbol{\phi}(\boldsymbol{u}_t) \otimes \mathbb{E}_{\mathbf{x}_t, \mathbf{x}_{t+1}}[x_t \mathbf{x}_{t+1}^\top]$$

The quantity of interest for the M-step from the complete data log-likelihood is (up to an additive constant)

$$\mathbb{E}\left[\sum_{t=1}^{T-1}\log p(\mathbf{x}_{t+1} \mid \mathbf{x}_t, \mathbf{F}, \boldsymbol{u}_t)\right]$$

$$= -\frac{1}{2}\mathbb{E}\left[\sum_{t=1}^{T-1}(\mathbf{x}_{t+1} - (\mathbf{A}(\boldsymbol{u}_t)\mathbf{x}_t + \mathbf{b}(\boldsymbol{u}_t)))^\top Q^{-1}(\mathbf{x}_{t+1} - (\mathbf{A}(\boldsymbol{u}_t)\mathbf{x}_t + \mathbf{b}(\boldsymbol{u}_t)))\right] + \text{const.}$$

$$= -\frac{1}{2}\mathbb{E}\left[\sum_{t=1}^{T-1}\mathbf{x}_{t+1}^\top Q^{-1}\mathbf{x}_{t+1} - 2\mathbf{x}_{t+1}^\top Q^{-1}\mathbf{A}(\boldsymbol{u}_t)\mathbf{x}_t - 2\mathbf{x}_{t+1}^\top Q^{-1}\mathbf{b}(\boldsymbol{u}_t)\right.$$

$$\left. + \mathbf{x}_t^\top \mathbf{A}(\boldsymbol{u}_t)^\top Q^{-1}\mathbf{A}(\boldsymbol{u}_t)\mathbf{x}_t + 2\mathbf{x}_t^\top \mathbf{A}(\boldsymbol{u}_t)^\top Q^{-1}\mathbf{b}(\boldsymbol{u}_t) + \mathbf{b}(\boldsymbol{u}_t)^\top Q^{-1}\mathbf{b}(\boldsymbol{u}_t)\right] + \text{const.}$$

$$= -\frac{1}{2}\text{Tr}\left[\sum_{t=1}^{T-1}Q^{-1}\mathbb{E}[\mathbf{x}_{t+1}\mathbf{x}_{t+1}^\top] - 2Q^{-1}\mathbf{A}(\boldsymbol{u}_t)\mathbb{E}[\mathbf{x}_t\mathbf{x}_{t+1}^\top] - 2Q^{-1}\mathbf{b}(\boldsymbol{u}_t)\mathbb{E}[\mathbf{x}_{t+1}^\top]\right.$$

$$\left. + \mathbf{A}(\boldsymbol{u}_t)^\top Q^{-1}\mathbf{A}(\boldsymbol{u}_t)\mathbb{E}[\mathbf{x}_t\mathbf{x}_t^\top] + 2\mathbf{A}(\boldsymbol{u}_t)^\top Q^{-1}\mathbf{b}(\boldsymbol{u}_t)\mathbb{E}[\mathbf{x}_t^\top] + \mathbf{b}(\boldsymbol{u}_t)^\top Q^{-1}\mathbf{b}(\boldsymbol{u}_t)\right] + \text{const.}$$

Which with the finite basis expansion reads as

$$\mathcal{L} = -\frac{1}{2}\text{Tr}\left[\sum_{t=1}^{T-1}Q^{-1}\mathbb{E}[\mathbf{x}_{t+1}\mathbf{x}_{t+1}^\top] - 2Q^{-1}\mathbf{W}_A^\top(\boldsymbol{\phi}(\boldsymbol{u}_t) \otimes \mathbb{E}[\mathbf{x}_t\mathbf{x}_{t+1}^\top]) - 2Q^{-1}\mathbf{W}_b^\top(\boldsymbol{\phi}(\boldsymbol{u}_t)\mathbb{E}[\mathbf{x}_{t+1}^\top])\right.$$

$$\left. + \mathbf{W}_A Q^{-1}\mathbf{W}_A^\top(\boldsymbol{\phi}(\boldsymbol{u}_t)\boldsymbol{\phi}(\boldsymbol{u}_t)^\top \otimes \mathbb{E}[\mathbf{x}_t\mathbf{x}_t^\top]) + 2\mathbf{W}_A Q^{-1}\mathbf{W}_b^\top(\boldsymbol{\phi}(\boldsymbol{u}_t)\boldsymbol{\phi}(\boldsymbol{u}_t)^\top \otimes \mathbb{E}[\mathbf{x}_t^\top])\right.$$

$$\left. + \mathbf{W}_b Q^{-1}\mathbf{W}_b^\top \boldsymbol{\phi}(\boldsymbol{u}_t)\boldsymbol{\phi}(\boldsymbol{u}_t)^\top\right]$$

The partial derivatives satisfy, denoting $\phi_t = \phi(\boldsymbol{u}_t)$,

$$\frac{\partial \mathcal{L}}{\partial \mathbf{W}_A} = \sum_{t=1}^{T-1} (\phi_t \otimes \mathbb{E}[\mathbf{x}_t \mathbf{x}_{t+1}^\top]) Q^{-1} - (\phi_t \phi_t^\top \otimes \mathbb{E}[\mathbf{x}_t \mathbf{x}_t^\top]) \mathbf{W}_A Q^{-1} - (\phi_t \phi_t^\top \otimes \mathbb{E}[\mathbf{x}_t]^\top)^\top \mathbf{W}_b Q^{-1}$$

$$=: N_\Delta Q^{-1} - N_{(1,T-1)} \mathbf{W}_A Q^{-1} - (\Phi^\top Z)^\top \mathbf{W}_b Q^{-1} \tag{14}$$

$$\frac{\partial \mathcal{L}}{\partial \mathbf{W}_b} = \sum_{t=1}^{T-1} \phi_t \mathbb{E}[\mathbf{x}_{t+1}^\top] Q^{-1} - (\phi_t \phi_t^\top \otimes \mathbb{E}[\mathbf{x}_t^\top]) \mathbf{W}_A Q^{-1} + \phi_t \phi_t^\top \mathbf{W}_b Q^{-1}$$

$$=: \Phi^\top X Q^{-1} - \Phi^\top Z \mathbf{W}_A Q^{-1} + \Phi^\top \Phi \mathbf{W}_b Q^{-1} \tag{15}$$

where we've defined the matrices $\Phi \in \mathbb{R}^{(T-1)\times L}$, $Z \in \mathbb{R}^{(T-1)\times DL}$, $X \in \mathbb{R}^{(T-1)\times D}$ obtained by stacking $\phi(\boldsymbol{u}_t)$, $\phi(\boldsymbol{u}_t) \otimes \mathbb{E}[\mathbf{x}_t]$ and $\mathbb{E}[\mathbf{x}_{t+1}]$ respectively for $t \in \{1, \ldots, T-1\}$, and the sufficient statistics

$$N_{(t_1,t_2)} = \sum_{t_1}^{t_2} \phi_t \phi_t^\top \otimes \mathbb{E}[\mathbf{x}_t \mathbf{x}_t^\top], \quad N_\Delta = \sum_{t=1}^{T-1} \phi(\boldsymbol{u}_t) \otimes \mathbb{E}[\mathbf{x}_t \mathbf{x}_{t+1}^\top].$$

which are all defined during our E-step. These statistics are in contrast to the "typical" sufficient stats without the weight-space parametrization

$$M_{(t_1,t_2)} = \sum_{t_1}^{t_2} \mathbb{E}[\mathbf{x}_t \mathbf{x}_t^\top], \quad M_\Delta = \sum_{t=1}^{T-1} \mathbb{E}[\mathbf{x}_t \mathbf{x}_{t+1}^\top].$$

Incorporating the Gaussian prior on $\mathbf{W}_A$ and $\mathbf{W}_b$ and equating the two partial derivatives in (14-15) to 0 to obtain the stationary points, we obtain the system of equations

$$\begin{bmatrix} \mathbf{W}_A \\ \mathbf{W}_b \end{bmatrix} Q + \begin{bmatrix} N_{(1,T-1)} & Z^\top \Phi \\ \Phi^\top Z & -\Phi^\top \Phi \end{bmatrix} \begin{bmatrix} \mathbf{W}_A \\ \mathbf{W}_b \end{bmatrix} = \begin{bmatrix} N_\Delta \\ \Phi^\top X \end{bmatrix} \tag{16}$$

which solving for $\mathbf{W}_A$ and $\mathbf{W}_b$ jointly amounts to solving our M-step.

## A.2 Nonlinear dynamics: linearization and composite dynamics

Consider input-driven nonlinear dynamics in $\mathbf{x}_t \in \mathbb{R}^D$,

$$\mathbf{x}_{t+1} = \boldsymbol{f}(\mathbf{x}_t, \mathbf{u}_t) + \epsilon_t \tag{17}$$

governed by $\boldsymbol{f} : \mathbb{R}^D \times \mathcal{U} \to \mathbb{R}^D$, and where $\epsilon_t$ is zero-mean Gaussian noise. We assume that $\boldsymbol{f}$ has continuous and bounded second-order partial derivatives in both $\mathbf{x}$ and $\mathbf{u}$. Below we treat the state and input variables $(\mathbf{x}_t, \mathbf{u}_t)$ as random variables that are jointly drawn from some unspecified distribution.

### A.2.1 CLDS conditional approximation error

Let $f_1, \ldots, f_D$ denote the output dimensions of $\boldsymbol{f}$; that is, $\boldsymbol{f}(\mathbf{x}_t, \mathbf{u}_t) = [f_1(\mathbf{x}_t, \mathbf{u}_t) \quad \ldots \quad f_D(\mathbf{x}_t, \mathbf{u}_t)]^\top$. In a first step to relate our CLDS dynamics to $\boldsymbol{f}$, consider the first-order Taylor expansion for each output dimension in the first argument $\boldsymbol{x}$ about $\boldsymbol{a} \in \mathbb{R}^D$. For $i \in \{1, \ldots, D\}$, this is

$$f_i(\mathbf{x}_t, \mathbf{u}_t) = f_i(\boldsymbol{a}, \mathbf{u}_t) + \nabla_{\boldsymbol{x}} f_i(\boldsymbol{a}, \mathbf{u}_t)^\top (\mathbf{x}_t - \boldsymbol{a}) + \mathcal{E}_i \tag{18}$$

where $\nabla_{\boldsymbol{x}} f_i$ denotes the vector-valued gradient of $f_i$ with respect to it's first argument $\boldsymbol{x}$ and $\mathcal{E}_i$ is the residual of the Taylor approximation. The Lagrange remainder form of Taylor's theorem tells us that this residual can be expressed as:

$$\mathcal{E}_i = (\mathbf{x}_t - \boldsymbol{a})^\top \nabla_{\boldsymbol{x}}^2 f_i(\zeta, \mathbf{u}_t)(\mathbf{x}_t - \boldsymbol{a}) \tag{19}$$

for some $\zeta \in \mathbb{R}^D$, where $\nabla_{\boldsymbol{x}}^2 f_i$ is the matrix-valued Hessian of $f_i$ with respect to its first argument. We can upper bound the absolute value of this remainder using the Cauchy-Schwartz inequality and a standard operator norm inequality. Specifically, for $i \in \{1, \ldots, D\}$, we have

$$|\mathcal{E}_i| \leq \left\| \nabla_{\boldsymbol{x}}^2 f_i(\zeta, \mathbf{u}_t) \right\|_2 \|\mathbf{x}_t - \boldsymbol{a}\|_2^2 \tag{20}$$

where $\left\|\nabla_{\boldsymbol{x}}^2 f_i(\zeta, \mathbf{u}_t)\right\|_2$ denotes the maximal singular value (operator norm) of the matrix $\nabla_{\boldsymbol{x}}^2 f_i(\zeta, \mathbf{u}_t) \in \mathbb{R}^{D \times D}$. We assume that this operator norm is upper bounded globally by a constant $L_i > 0$,

$$\left\|\nabla_{\boldsymbol{x}}^2 f_i(\mathbf{x}, \mathbf{u})\right\|_2 \leq L_i \qquad \forall\, \mathbf{x}, \mathbf{u} \in \mathbb{R}^D. \tag{21}$$

Intuitively, this assumption implies that the second-order derivatives of $\boldsymbol{f}$ with respect to $\mathbf{x}$ are not too large, meaning that the accuracy of the first-order Taylor approximation degrades in proportion to the magnitude of curvature in $\boldsymbol{f}$.

Returning to equation (20), we proceed by taking conditional expectations with respect to $\mathbf{x}_t$ given $\mathbf{u}_t$ on both sides of the inequality. This yields an upper bound on the expected approximation error,

$$\mathbb{E}\left[|\mathcal{E}_i| \mid \mathbf{u}_t\right] \leq L_i \cdot \mathbb{E}\left[\left.\|\mathbf{x}_t - \boldsymbol{a}\|_2^2 \,\right|\, \mathbf{u}_t\right].$$

This upper bound is minimized by choosing $\boldsymbol{a} = \mathbb{E}[\mathbf{x}_t \mid \mathbf{u}_t]$. Plugging in this choice, we observe that

$$\mathbb{E}\left[|\mathcal{E}_i| \mid \mathbf{u}_t\right] \leq L_i \cdot \mathrm{Tr}\left[\mathbb{C}\mathrm{ov}[\mathbf{x}_t \mid \mathbf{u}_t]\right]$$

where $\mathbb{C}\mathrm{ov}[\mathbf{x}_t \mid \mathbf{u}_t] = \mathbb{E}\left[(\mathbf{x}_t - \mathbb{E}[\mathbf{x}_t \mid \mathbf{u}_t])(\mathbf{x}_t - \mathbb{E}[\mathbf{x}_t \mid \mathbf{u}_t])^\top \mid \mathbf{u}_t\right]$ is the conditional covariance of $\mathbf{x}_t$ given $\mathbf{u}_t$. Finally, we can sum these upper bounds over $i = \{1, \dots, D\}$ to bound the total expected approximation error as

$$\sum_i \mathbb{E}\left[|\mathcal{E}_i| \mid \mathbf{u}_t\right] \leq L \cdot \mathrm{Tr}\left[\mathbb{C}\mathrm{ov}[\mathbf{x}_t \mid \mathbf{u}_t]\right] \tag{22}$$

where we have defined $L = \sum_i L_i$ as a global constant bounding the second-order smoothness of $\boldsymbol{f}$ across all dimensions.

Returning to equation (18) and plugging in the optimal choice of $\mathbf{a} = \mathbb{E}[\mathbf{x}_t \mid \mathbf{u}_t]$, we obtain the following CLDS approximation to the nonlinear dynamics

$$\boldsymbol{h}(\mathbf{x}_t, \mathbf{u}_t) := \boldsymbol{f}(\mathbb{E}[\mathbf{x}_t \mid \mathbf{u}_t], \mathbf{u}_t) + \nabla_{\boldsymbol{x}}\boldsymbol{f}(\mathbb{E}[\mathbf{x}_t \mid \mathbf{u}_t], \mathbf{u}_t)(\mathbf{x}_t - \mathbb{E}[\mathbf{x}_t \mid \mathbf{u}_t]) = \boldsymbol{A}(\mathbf{u}_t)\mathbf{x}_t + \boldsymbol{b}(\mathbf{u}_t) \tag{23}$$

where $\nabla_{\boldsymbol{x}}\boldsymbol{f}$ is the matrix-valued Jacobian, $\nabla_{\boldsymbol{x}}\boldsymbol{f}(\boldsymbol{x}, \boldsymbol{u}) \in \mathbb{R}^{D \times D}$, with respect to the first argument of $\boldsymbol{f}$ and we have re-arranged the terms and defined

$$\boldsymbol{A}(\mathbf{u}_t) := \nabla_{\boldsymbol{x}}\boldsymbol{f}(\mathbb{E}[\mathbf{x}_t \mid \mathbf{u}_t], \mathbf{u}_t), \quad \boldsymbol{b}(\mathbf{u}_t) := \boldsymbol{f}(\mathbb{E}[\mathbf{x}_t \mid \mathbf{u}_t], \mathbf{u}_t) - \nabla_{\boldsymbol{x}}\boldsymbol{f}(\mathbb{E}[\mathbf{x}_t \mid \mathbf{u}_t], \mathbf{u}_t)\mathbb{E}[\mathbf{x}_t \mid \mathbf{u}_t].$$

For each value of $\mathbf{u}_t$, the quality of this approximation is guaranteed by equation (22) to be small if the second-order derivatives of $\boldsymbol{f}$ with respect to $\mathbf{x}$ are small and if the conditional variance of $\mathbf{x}_t$ given $\mathbf{u}_t$ is small. We note that this analysis of approximation error and does not account for the additional estimation error we incur when learning the functions $\boldsymbol{A}(\mathbf{u})$ and $\boldsymbol{b}(\mathbf{u})$ from noisy and limited data. Nonetheless, this analysis tells us that we expect the CLDS to perform well in circumstances where the underlying dynamics are smooth and the conditional distributions of $\mathbf{x}_t$ given $\mathbf{u}_t$ have low variance.

### A.2.2 Composite dynamics

When considering the *composite dynamics* in (7) (Section §3.1), we are interested in approximating the input-driven nonlinear dynamics given by equation (17) with autonomous nonlinear dynamics, governed by some function $\boldsymbol{g}$ such that

$$\boldsymbol{f}(\mathbf{x}_t, \mathbf{u}_t) \approx \boldsymbol{g}(\mathbf{x}_t).$$

We can evaluate the quality of this approximation using the conditional expectation of the squared error

$$\mathbb{E}\left[\|\boldsymbol{f}(\mathbf{x}_t, \mathbf{u}_t) - \boldsymbol{g}(\mathbf{x}_t)\|_2^2 \,\big|\, \mathbf{x}_t\right]$$

where the conditional expectation is taken over $\mathbf{u}_t$ given $\mathbf{x}_t$. This approximation error is minimized by choosing

$$\boldsymbol{g}(\mathbf{x}_t) = \mathbb{E}[\boldsymbol{f}(\mathbf{x}_t, \mathbf{u}_t) \mid \mathbf{x}_t] = \mathbb{E}[\mathbf{x}_{t+1} \mid \mathbf{x}_t]$$

However, learning $\boldsymbol{f}$ from limited data is challenging, so we replace this with our CLDS model to achieve the composite dynamical system

$$\boldsymbol{g}(\mathbf{x}_t) \approx \mathbb{E}[\boldsymbol{h}(\mathbf{x}_t, \mathbf{u}_t) \mid \mathbf{x}_t] \tag{24}$$

where $h(\mathbf{x}, \mathbf{u}) = A(\mathbf{u})\mathbf{x} + b(\mathbf{u})$ as in equation (23). Now we analyze the quality of this approximation. Consider the residual along dimension $i \in \{1, \dots, D\}$,

$$\mathcal{R}_i = f_i(\mathbf{x}_t, \mathbf{u}_t) - \mathbb{E}[h_i(\mathbf{x}_t, \tilde{\mathbf{u}}_t) \mid \mathbf{x}_t]$$

where the expectation in the second term is taken over $\tilde{\mathbf{u}}_t$, which is drawn from the conditional distribution of $\mathbf{u}_t$ given $\mathbf{x}_t$. That is, $\mathcal{R}_i$ is a random variable that depends on a joint sample of $(\mathbf{x}_t, \mathbf{u}_t)$ from the stationary distribution, and $\tilde{\mathbf{u}}_t$ is a dummy variable that is integrated out during the calculation of $\mathcal{R}_i$. To proceed, we note that

$$\begin{aligned}
\mathcal{R}_i &= \mathbb{E}[f_i(\mathbf{x}_t, \mathbf{u}_t) - h_i(\mathbf{x}_t, \tilde{\mathbf{u}}_t) \mid \mathbf{x}_t] \\
&= \mathbb{E}[f_i(\mathbf{x}_t, \mathbf{u}_t) - f_i(\mathbf{x}_t, \tilde{\mathbf{u}}_t) + f_i(\mathbf{x}_t, \tilde{\mathbf{u}}_t) - h_i(\mathbf{x}_t, \tilde{\mathbf{u}}_t) \mid \mathbf{x}_t]
\end{aligned}$$

and, applying Jensen's inequality and the triangle inequality, we conclude

$$\begin{aligned}
|\mathcal{R}_i| &\leq \mathbb{E}_{\tilde{\mathbf{u}}_t|\mathbf{x}_t}\big[\big|f_i(\mathbf{x}_t, \mathbf{u}_t) - f_i(\mathbf{x}_t, \tilde{\mathbf{u}}_t) + f_i(\mathbf{x}_t, \tilde{\mathbf{u}}_t) - h_i(\mathbf{x}_t, \tilde{\mathbf{u}}_t)\big|\big] \\
&\leq \mathbb{E}_{\tilde{\mathbf{u}}_t|\mathbf{x}_t}\big[\big|f_i(\mathbf{x}_t, \mathbf{u}_t) - f_i(\mathbf{x}_t, \tilde{\mathbf{u}}_t)\big|\big] + \mathbb{E}_{\tilde{\mathbf{u}}_t|\mathbf{x}_t}\big[\big|f_i(\mathbf{x}_t, \tilde{\mathbf{u}}_t) - h_i(\mathbf{x}_t, \tilde{\mathbf{u}}_t)\big|\big]
\end{aligned}$$

where we have introduced a minor change in notation, using $\mathbb{E}_{\tilde{\mathbf{u}}_t|\mathbf{x}_t}$ to denote the conditional expectation of $\tilde{\mathbf{u}}_t$, given $\mathbf{x}_t$. Now we take expectations on both sides of this inequality with respect to the remaining random variables, $\mathbf{x}_t$ and $\mathbf{u}_t$, which are sampled from some stationary distribution associated with the dynamical system. Using the law of total expectation, we obtain

$$\mathbb{E}|\mathcal{R}_i| \leq \mathbb{E}_{\mathbf{x}_t}\big[\mathbb{E}_{\mathbf{u}_t, \tilde{\mathbf{u}}_t|\mathbf{x}_t}\big[\big|f_i(\mathbf{x}_t, \mathbf{u}_t) - f_i(\mathbf{x}_t, \tilde{\mathbf{u}}_t)\big|\big]\big] + \mathbb{E}_{\mathbf{u}_t}\big[\mathbb{E}_{\mathbf{x}_t|\mathbf{u}_t}\big[\big|f_i(\mathbf{x}_t, \mathbf{u}_t) - h_i(\mathbf{x}_t, \mathbf{u}_t)\big|\big]\big] \tag{25}$$

On the right hand side, the first term takes the conditional expectation over $\mathbf{u}_t$ and $\tilde{\mathbf{u}}_t$, followed by an expectation over $\mathbf{x}_t$. The second term reverses this order, taking the conditional expectation over $\mathbf{x}_t$, followed by an expectation over $\mathbf{u}_t$ (since this is identically distributed to $\tilde{\mathbf{u}}_t$, we drop the tilde).

To upper bound the first term, we introduce an assumption that $f_i$ is Lipschitz continuous in its second argument. That is, there exists a constant $C_i > 0$ such that

$$|f_i(\mathbf{x}_t, \mathbf{u}) - f_i(\mathbf{x}_t, \mathbf{u}')| \leq C_i \|\mathbf{u} - \mathbf{u}'\|_2 \qquad \forall \mathbf{u}, \mathbf{u}' \in \mathcal{U}.$$

In conjunction with Jensen's inequality, this Lipschitz assumption implies the following upper bound:

$$\begin{aligned}
\mathbb{E}_{\mathbf{u}_t, \tilde{\mathbf{u}}_t|\mathbf{x}_t}\big[\big|f_i(\mathbf{x}_t, \mathbf{u}_t) - f_i(\mathbf{x}_t, \tilde{\mathbf{u}}_t)\big|\big] &\leq C_i \cdot \mathbb{E}_{\mathbf{u}_t, \tilde{\mathbf{u}}_t|\mathbf{x}_t}\big[\big\|\mathbf{u}_t - \tilde{\mathbf{u}}_t\big\|_2\big] \\
&\leq C_i \sqrt{\mathbb{E}_{\mathbf{u}_t, \tilde{\mathbf{u}}_t|\mathbf{x}_t}\big\|\mathbf{u}_t - \tilde{\mathbf{u}}_t\big\|_2^2} \\
&= C_i \sqrt{2\,\mathrm{Tr}[\mathbb{C}\mathrm{ov}[\mathbf{u}_t \mid \mathbf{x}_t]]}
\end{aligned}$$

It remains to upper bound the second term in equation (25). A direct application of the results in §A.2.1 yields the bound

$$\mathbb{E}_{\mathbf{x}_t|\mathbf{u}_t}\big[\big|f_i(\mathbf{x}_t, \mathbf{u}_t) - h_i(\mathbf{x}_t, \mathbf{u}_t)\big|\big] \leq L_i \cdot \mathrm{Tr}[\mathbb{C}\mathrm{ov}[\mathbf{x}_t \mid \mathbf{u}_t]]$$

where $L_i$, defined in (21), is a constant bounding the second derivatives of $f_i$ with respect to $\mathbf{x}_t$. Putting these pieces together we conclude that

$$\mathbb{E}|\mathcal{R}_i| \leq C_i \cdot \mathbb{E}_{\mathbf{x}_t} \sqrt{2\,\mathrm{Tr}[\mathbb{C}\mathrm{ov}[\mathbf{u}_t \mid \mathbf{x}_t]]} + L_i \cdot \mathbb{E}_{\mathbf{u}_t} \mathrm{Tr}[\mathbb{C}\mathrm{ov}[\mathbf{x}_t \mid \mathbf{u}_t]]$$

And so an upper bound on the total absolute error of the composite dynamics is given by

$$\sum_{i=1}^{D} \mathbb{E}|\mathcal{R}_i| \leq C \cdot \mathbb{E}_{\mathbf{x}_t} \sqrt{2\,\mathrm{Tr}[\mathbb{C}\mathrm{ov}[\mathbf{u}_t \mid \mathbf{x}_t]]} + L \cdot \mathbb{E}_{\mathbf{u}_t} \mathrm{Tr}[\mathbb{C}\mathrm{ov}[\mathbf{x}_t \mid \mathbf{u}_t]]$$

where we have defined $C = \sum_i C_i$ and $L = \sum_i L_i$.

In summary, we have shown that the approximation error of the composite dynamical system, defined in equation (7), is bounded by a sum of two terms. The first term approaches zero in the limit that the conditional covariance of $\mathbf{u}_t$ given $\mathbf{x}_t$ goes to zero, while the second term approaches zero in the limit that the conditional covariance of $\mathbf{x}_t$ given $\mathbf{u}_t$ goes to zero. Thus, the composite dynamics have the potential to provide an accurate depiction of the true nonlinear dynamical system if $\mathbf{x}_t$ and $\mathbf{u}_t$ are close to being in one-to-one correspondence with each other.

We note that in practice when computing the composite dynamics in (7), we make the simplifying assumption that $p(\mathbf{u}_t|\mathbf{x}_t = x)$ does not depend on $t$ (i.e. we assume that this decoding distribution is stationary).

### A.3 Correspondence between CLDS and Switching LDS

In this section, we explore the relationship between the linear time-variant dynamics of (1a) with $\mathbf{A}_{ij} \overset{iid}{\sim} \mathcal{GP}(0, k_t)$, and the dynamics of a switching linear dynamical system. While the latter has parameters evolving over a discrete set, we can nonetheless explore how to think of this discrete support as embedded within $\mathbb{R}^{D \times D}$, and seek to match the moments of these two processes.

The first thing to note is that by drawing the entries of $\mathbf{A}_{ij}$ i.i.d., we can gain insight by considering a single process $a(\cdot) \sim \mathcal{GP}(0, k_t)$, and a SLDS with one-dimensional dynamics. Thus, we consider the SLDS model

$$p(\mathbf{z}_{t+1} = i \mid \mathbf{z}_t = j) = P_{ij} \tag{26a}$$

$$\mathbf{x}_{t+1} = \mathbf{a}^{(\mathbf{z}_t)} \mathbf{x}_t + \epsilon_t \tag{26b}$$

with discrete states $\mathbf{z}_t$ governing dynamics in $\mathbf{x}_t$, with transition matrix $P$ and dynamics $\mathbf{a}^{(z)}$ for $z \in \mathcal{Z}$. Denote $\boldsymbol{z} = (z_1, \ldots, z_K)^\top \in \mathbb{R}^{|\mathcal{Z}|}$ for the vector of the $|\mathcal{Z}| = K$ values that can be taken by the stochastic process z, and similarly $\boldsymbol{a} = (a^{(z_1)}, \ldots, a^{(z_K)})^\top$. We consider $\pi$ the stationary distribution for the z process. Finally, note that the map $z_n \to a^{(z_n)}$ is deterministic, one-to-one and onto, such that we can think of the Markov chain in $\mathbf{z}_t$ as having support over $\boldsymbol{a}$. Let $\mathbf{a}_t := a^{(\mathbf{z}_t)}$, and denote $a_i = a^{(z_i)}$.

Let us now explore the moments of the stochastic process $\mathbf{a}_t$ to determine its relationship with the CLDS. Assume $\mathbf{z}_t$ has reached stationarity, such that $p(\mathbf{z}_t) = p(\mathbf{a}_t) = \pi$. Then, first,

$$\mathbb{E}[\mathbf{a}] = \pi^\top \boldsymbol{a} \tag{27}$$

and then we have the cross-correlation

$$\begin{aligned}
\mathbb{E}\left[\mathbf{a}_t \mathbf{a}_{t+n}\right] &= \mathbb{E}\left[\mathbb{E}\left[\mathbf{a}_t \mathbf{a}_{t+n} \mid \mathbf{a}_t\right]\right] \\
&= \sum_{j=1}^{K} \mathbb{E}\left[\mathbf{a}_t \mathbf{a}_{t+n} \mid \mathbf{a}_t = a_j\right] p(\mathbf{a}_t = a_j) \\
&= \sum_{j=1}^{K} \sum_{i=1}^{K} a_j \left(p(\mathbf{a}_{t+n} = z_i \mid \mathbf{a}_t = a_j)\right) p(\mathbf{a}_t = a_j) \\
&= \sum_{i=1}^{K} \sum_{j=1}^{K} a_i a_j P_{ij}^n \pi_j \\
&= \boldsymbol{a}^\top P^n \mathrm{diag}(\pi) \boldsymbol{a}.
\end{aligned}$$

Denote $\Pi = \mathrm{diag}(\pi)$. We get the desired covariance

$$\mathbb{C}\mathrm{ov}(\mathbf{a}_t, \mathbf{a}_{t+n}) = \mathbb{E}\left[\mathbf{a}_t \mathbf{a}_{t+n}\right] - \mathbb{E}\left[\mathbf{a}_t\right] \mathbb{E}\left[\mathbf{a}_{t+n}\right] = \boldsymbol{a}^\top \left(P^n \Pi - \pi \pi^\top\right) \boldsymbol{a},$$

yielding our kernel form

$$\mathbb{C}\mathrm{ov}(\mathbf{a}(t_i), \mathbf{a}(t_j)) = k_t(t_i, t_j) = \boldsymbol{a}^\top \left(P^{|t_j - t_i|} \Pi - \pi \pi^\top\right) \boldsymbol{a}. \tag{28}$$

Hence, in all, our approximation of the stochastic process $\mathbf{a}_t$ over $\mathbb{R}$ up to the first two moments is

$$\mathbf{a}(\cdot) \sim \mathcal{GP}\left(\pi^\top \boldsymbol{a}, \boldsymbol{a}^\top \left(P^{|t_j - t_i|} \Pi - \pi \pi^\top\right) \boldsymbol{a}\right) \tag{29}$$

which in particular can be made zero-mean by consider values $\boldsymbol{a}$ such that $\pi^\top \boldsymbol{a} = 0$. This establishes the form of the GP prior over $a(\cdot)$ for which the corresponding CLDS best matches the SLDS.

## B Experiments

### B.1 Model implementations

We initialize observation matrices for all models ($C$ in LDS, CLDS and gpSLDS models, log-rate decoder weights in LFADS) as the PCA principal axes in $\mathbf{y}$-space—that is, the top $D$ right singular vectors of the data—for each task.

**CLDS**  In all experiments, we assume that $\mathbf{d}(\boldsymbol{u}_t) = \mathbf{0}$, which forces the predicted firing rates, conditioned on $\boldsymbol{u}_t$, to lie in a $N$-dimensional space spanned by the columns of $\mathbf{C}(\boldsymbol{u}_t)$.

**gpSLDS**  We implemented the gpSLDS [19] using its accompanying code at `https://github.com/lindermanlab/gpslds`. We used a latent dimension of $D = 2$ in all experiments as we encountered significant OOM errors with higher dimensions. We used exactly the same inputs $\boldsymbol{u}_{1:T}$ as those provided to the (C)LDS models. We use $J = 2$ and the feature transformation

$$\boldsymbol{\phi}(\boldsymbol{x}) = \begin{bmatrix} 1 & \boldsymbol{x}_1 & \boldsymbol{x}_2 & \boldsymbol{x}_1^2 & \boldsymbol{x}_2^2 \end{bmatrix}^\top \tag{30}$$

as to allow both linear (as in the rSLDS) and circular $\boldsymbol{\pi}(\boldsymbol{x})$ partitions. We experimented with providing the same basis functions for the feature transformation as the ones provided to the CLDS but found that it did not provide any advantage. Other hyper-parameters we set to default provided values.

**LFADS**  We use the Jax implementation of LFADS available at `https://github.com/google-research/computation-thru-dynamics/tree/master/lfads_tutorial`, which is under a Apache-2.0 license. We choose the factor dimension to be the same as the latent dimension $D$ of the (C)LDS models and the inferred inputs to be of dimension $|\mathcal{U}|$, both following the (C)LDS models on any given experiment. The other components of the architecture are held fixed across all experiments:

- Encoder, controller and generator have hidden-states of dimension 32;
- The inferred inputs are modeled as having an auto-correlation of $1.0$ and a variance of $0.1$;
- We train for 1000 epochs, with an initial learning rate of $0.5$ with exponential decay at rate $0.995$, along with a KL warm-up coming in at 500 steps.

## B.2  Synthetic experiment and parameter recovery

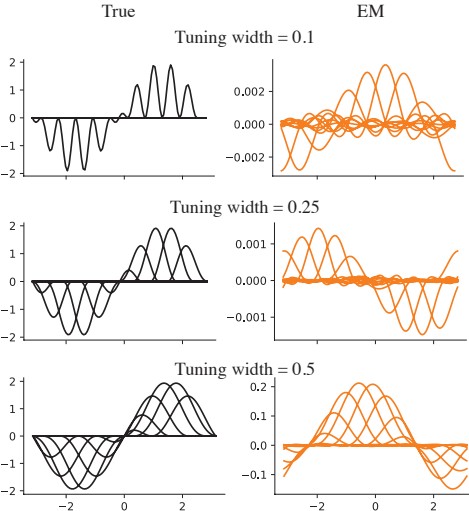

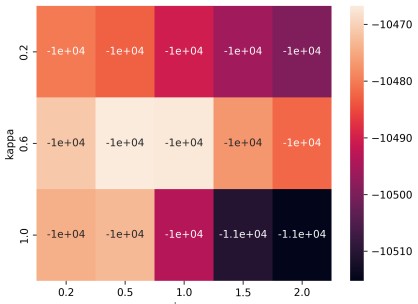

Figure 5: Recovery of $\mathbf{C}$. Rows indicate varying level of tuning curve width $\gamma$ for $\mathbf{C}_{i,:}(\boldsymbol{u})$. Recovery becomes more challenging for smaller width since it requires a higher and higher number of bases $L$ to approximate the true tuning bump.

Figure 6: Hyperparameter search, CLDS marginal log-likelihood on a held-out validation dataset set for $\kappa \in \{0.2, 0.6, 1.0\}$ and $\sigma \in \{0.2, 0.5, 1.0, 1.5, 2.0\}$. Maximum attained at $\{\kappa, \sigma\} = \{0.6, 0.5\}$.

For the peaks $\xi_i$ spanning regularly the interval $[-\pi, \pi)$ and widths $\gamma$, the tuning curves in the rows of $\mathbf{C}$ for the synthetic experiment are defined as

$$\boldsymbol{C}_{i,:}(\boldsymbol{u}) = \begin{cases} \left(1 + \cos\left(\frac{\boldsymbol{u} - \xi_i}{\gamma}\right)\right) \boldsymbol{u}^\top & \text{if } \boldsymbol{u} \in (\xi_i - \gamma\pi, \xi_i + \gamma\pi) \\ 0 & \text{else} \end{cases} \tag{31}$$

We plot the its recovery with our inference procedure in Fig. 5, up to the invertible transform non-identifiability inherent to LDS models.

In Table 1, we report the results of our inference method on the first synthetic ring-attractor experiment (§3.2) for varying emission noise scale $R = \sigma_R^2 I$, keeping all other experimental setup parameters exactly the same as in the original experiment and fixed. We see that even as the signal-to-noise ratio degrades and we reach a co-smoothing value of $0.21$, which is lower than our fit on the monkey data of §3.5, we still have decent $\mathbf{A}$ recovery and near-perfect $R$ recovery.

| **True noise log scale** $\log \sigma_R$ | $-2$ | $-1$ | $0$ | $1$ |
|---|---|---|---|---|
| Recovered $R$ log scale | $-1.97$ | $-0.98$ | $0.02$ | $1.02$ |
| $\mathbf{A}(\cdot)$ recovery error | $0.01$ | $0.02$ | $0.11$ | $0.32$ |
| Co-smoothing $R^2$ | $0.99$ | $0.94$ | $0.68$ | $0.21$ |

Table 1: *Impact of observational noise on quality of fit.* The "recovered $R$ scale" is the square-root of the matrix 2-norm of $R$. The $\mathbf{A}(\cdot)$ recovery error is the average in 2-norm error (between recovered and true) in eigenvalues over a regular grid of 50 angular conditions $\theta$. The co-smoothing $R^2$ is the average validation set $R^2$ on top-5-variance neurons, like in the main text.

## B.3   Neural circuit ring attractor experiment

Let $y(\phi, t)$ be the firing-rate "bump" on angles $\phi \in [0, 2\pi)$ with periodic boundary conditions, and let $v(t) = \dot{\theta}(t)$ be the angular velocity derived from the head-direction input. We consider its dynamics governed by the stochastic PDE

$$dy(\phi, t) = \tau \left[ -y(\phi, t) + f\big((w * y)(\phi, t)\big) + \kappa_v v(t)\, \partial_\phi y(\phi, t) \right] dt + \sigma dB(\phi, t)$$

with

- $f(\cdot)$ pointwise ReLU nonlinearity.
- The recurrent drive is a circular convolution

$$(w * y)(\phi, t) \;=\; \int_0^{2\pi} w(\phi - \psi)\, y(\psi, t)\, d\psi,$$

  with a rotationally symmetric kernel (in radians)

$$W(\Delta) = J_e \exp\left(-\tfrac{1}{2}(\Delta/\sigma_e)^2\right) - J_i \exp\left(-\tfrac{1}{2}(\Delta/\sigma_i)^2\right)$$

  for $\Delta \in (-\pi, \pi]$ (wrapped), $\{J_e, \sigma_e\}$, $\{J_i, \sigma_i\}$ parameters of excitation and inhibition.
- The term $\kappa_v\, v(t)\, \partial_\phi y$ shifts the bump at speed proportional to $v(t)$, of parameter $\kappa_v$.
- $B$ cylindrical Wiener process on the ring with covariance $\mathbb{E}[B(\phi, t)\, B(\phi', t')] = \delta(\phi - \phi')\, \min(t, t')$ ("white-noise" $dB/dt$).
- $\tau > 0$ a the time constant.

**Numerical implementation**   We discretize time to $\Delta t$ and head-direction to $\phi_i$, $i \in \{1, \ldots, N\}$, $N = 32$, regularly spanning $[0, 2\pi)$. Let $y(\phi_i, n\Delta t) = [\mathbf{y}_n]_i$. Consider $W$ a circulant matrix implementing $w$, and $D$ a centered difference discretization of the derivative. Our update is

$$\dot{\mathbf{y}}_t = -\mathbf{y}_t + f(W \mathbf{y}_t) + \kappa_v\, \theta_t\, D \mathbf{y}_t$$
$$\mathbf{y}_{t+1} = \mathbf{y}_t + \tau\, \Delta t\, \dot{\mathbf{y}}_t + \sigma\, \sqrt{\Delta t}\, \eta, \quad \eta \sim \mathcal{N}(0, 1)$$

as the Euler–Maruyama discretization of the SPDE.

After a search over kernel hyperparameters, we found the best-performing model (Fig. 7) to encode a ring of fixed points that closely resembles the synthetic HD fixed points (§3.2), and with eigenvalues closer to the ones observed from the CLDS fits on the mouse ADn recordings (§3.4). The results make us confident that the CLDS can capture this nonlinear structure even under model mismatch.

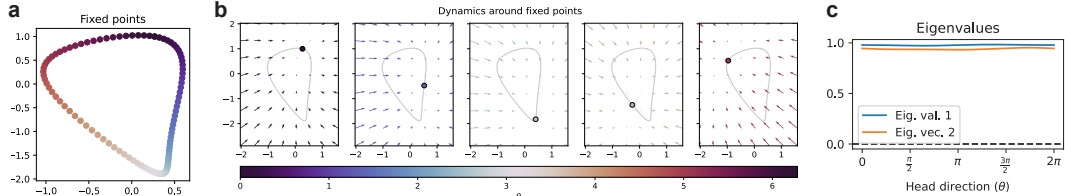

Figure 7: Inferred CLDS fixed points (**a**), local dynamics (**b**) and eigenvalues of **A** (**c**) as a function of head-direction $\theta$ for the neural circuit ring attractor experiment. Hyperparameter values are $\{\sigma, \kappa\} = \{1.0, 0.4\}$, obtained held-out LL over a grid search.

## B.4 Data and Pre-processing

**Mice Head-Direction**  We analyze neural recordings of antero-dorsal thalamic nucleus (ADn) from Ref. [32] of the mouse HD system in mice foraging in an open environment (Fig. 3**a**). The data was accessed through the `pynapple` (PYthon Neural Analysis Package, `https://pynapple.org/`, MIT License) package, with the data itself stored in NWB format on OSF at `https://osf.io/jb2gd`.

We considered neural activity from the "wake" period, binned in 50ms time-bins, then processed to firing rates by smoothing over a window of 4 bins, and separated into 10s trials.

**Macaque center-out reaching**  We analyzed neural recordings of dorsal premotor cortex (PMd) in macaques performing center-out reaching task from Ref. [33]—the associated article is under a CC BY 4.0 license.

The experiments supported 3 different reach radii, and we selected only the middle reach radius at 8cm. We were left with only the angular direction as the reach condition, over $N = 16$ possible reach angles. We aligned all trials around the go-cue, selecting 200 ms before the go-cue and 300ms after. We binned the data into 5ms bins, and performed Gaussian kernel smoothing with a standard deviation of 0.5 over bins.

## B.5 Hyper-parameters

Throughout all experiments, we've set $L = 5$ to balance expressivity and number of parameters. While it is possible to treat $L$ as a hyperparameter tuned to improve performance, we propose to instead set $L$ to a large enough value that results in negligible error in the GP kernel approximation. As $L \to \infty$, we pay a larger computational price in fitting the model, but the statistical properties of the model are essentially unchanged because high-frequency basis functions have nearly zero amplitude. This idea is well established in GP regression literature [56].

To select the other hyper-parameters of GP prior length-scale $\kappa$ and scale $\sigma$, we evaluated the various models on a held-out validation set. We plot in Fig. 6 our search over $\{\kappa, \sigma\}$ on the macaque center-out reaching data.

Finally, a dimensionality of $D = 2$ for the head-direction experiments is chosen throughout for interpretability, as the latent space is thought to encode head-direction. For the macaque center-out reaching data, we computed co-smoothing over $D \in \{3, 5, 10, 15\}$ (Table 2) and selected $D = 5$.

| $D$ | 3 | 5 | 10 | 15 |
|------|-------|-------|-------|-------|
| CLDS | 0.229 | 0.232 | 0.207 | 0.168 |
| LDS  | 0.211 | 0.193 | 0.196 | 0.170 |

Table 2: Co-smoothing $R^2$ reconstruction of single held-out neurons from the test-set, over varying latent dimensionality $D$, averaged over two random seeds.

## B.6 Compute Resources

We list below the compute resources used per experiment:

1. **Synthetic HD experiment**: Results were computed on a personal machine (Apple MacBook Pro, M2 Pro chip, 32G RAM). Inference converges in a few seconds.

2. **Mouse HD experiment**: Results were computed on a personal machine (Apple MacBook Pro, M2 Pro chip, 32G RAM). Inference converges in a few minutes.

3. **Monkey-reaching experiment**: Results were computed on an external cluster equipped with NVIDIA A100 GPUs. Fitting a single model typically used 60GB of GPU memory. All CLDS models were run strictly on CPU-only nodes (Intel Xeon Silver 4309Y Processor, 2.80 GHz, 8 cores/16 threads), using approximately 2 GB of RAM per run. Wall-clock times were between ten minutes and one hour.

