# OpenReview forum: "Modeling Neural Activity with Conditionally Linear Dynamical Systems"
_NeurIPS.cc/2025/Conference — NeurIPS 2025 poster_

### Official Review · Reviewer_hqD4 · 2025-06-17

**Clarity:** 3
**Significance:** 3
**Originality:** 3
**Rating:** 5
**Confidence:** 4

**Summary:**

The authors introduce conditionally linear dynamical systems (CLDS), where system parameters are modulated nonlinearly as a function of observed covariates $u_t$ (e.g., via a kernel mapping). This allows the latent dynamics to flexibly vary across contexts or conditions, while retaining the interpretability and tractability of linear state-space models. The model is fit using an approximate GP prior over parameters, enabling closed-form EM updates under Gaussian noise assumptions.

**Questions:**

*Comparisons/Related Work*:
The proposed approach shares a lot of conceptual similarities linear/structured SSMs, particularly to Mamba, where the recurrence kernel is conditioned on the input sequence, yielding a similarly dynamically reparameterized linear system. While the technical motivations differ (e.g., sequence modeling vs. spike dynamics), it could be beneficial to include this in the related work discussion, also with respect to training algorithms, benefits of linearity etc. Another recent line of work (Almost-Linear RNNs, Brenner et al., NeurIPS 2024) bears close conceptual similarity to recurrent switching linear DS, and could be mentioned.

**Ethical Concerns:**

["NO or VERY MINOR ethics concerns only"]

**Final Justification:**

An overall interesting and meaningful contribution. I already had a positive impression of the paper. The authors mostly adressed my concerns my concerns with novel experiments and clarifications. One small remaining open point I raised (a different noise model) is not an integral part of this paper's contribution, and does not preclude me from voting for acceptance.

**Limitations:**

Clear limitations section.

**Quality:**

3

**Strengths And Weaknesses:**

*Strengths*:
Overall, I like this paper. It’s clearly written, easy to follow, and communicates its ideas well. The motivation for combining input-dependent parameter modulation with linear dynamics is compelling, especially in neuroscience where interpretability matters. The figures are informative, and the structure of the paper makes it straightforward to trace the reasoning from model definition to inference to experiments. The theoretical results in Appendix A.2, especially the bounds on the approximation error for composite dynamics (A.2.2), are a valuable addition. Overall, it’s a useful contribution that sits in an important design space between black-box flexibility and mechanistic transparency.

*Weaknesses*:

*Overall Weakness*:
 While the paper presents a clean and interpretable framework, its empirical scope feels slightly narrow, and the evaluation does not seem to probe the limits or broader applicability of the method. All experiments are tailored to idealized conditions that align well with the model’s assumptions (Gaussian observations, well-structured modulators, modest data scale, and low latent dimensionality). Many of the paper’s selling points (e.g., “easy inference,” “simple extensions,” interpretability) are stated as general advantages, but it feels like they are not properly stress-tested or fully explored in more challenging settings. This leaves open the question of whether the performance gains and interpretability benefits are worth the constraints introduced by the modeling assumptions.

That said, if the authors are able to convincingly address these concerns, e.g. through additional ablations, clarifications of model limitations, and a deeper discussion of scalability and generalizability, I would be happy to support acceptance.

*Spike Data*:
The authors emphasize that one of the key advantages of CLDS models is ease of inference, particularly under a Gaussian noise model where Kalman smoothing and closed-form EM enable efficient learning (§2.3). They also note that the extension to non-Gaussian observations (e.g., Poisson spike counts) is straightforward in principle, since the posterior remains log-concave and tractable. However, all real data applications, including the macaque spike recordings, still use the Gaussian observation model. While this is mentioned in the limitations, this feels like a missed opportunity, particularly since the whole paper is tailored to "modeling neural activity": if extending to count-based models is practical and aligns better with the data-generating process, it would have been useful to demonstrate this empirically. At minimum, a concrete discussion of what would be required to implement this extension (and what challenges, if any, prevented it here) would help clarify how straightforward this path really is. As it stands, the claim of “easy extensibility” feels somewhat contradicted by its absence in the main experiments, where it would fit perfectly.

*EM Scaling*:
As a related point (given EM is highlighted as a major strength of the CLDS framework) while EM is indeed attractive for small- to medium-scale models, it is well known to scale badly with increasing latent dimensionality or large datasets (which would become relevant in many modern neural recording settings). The authors mention that “in principle” the model could be extended to use variational inference or MCMC methods, but do not test these alternatives or provide a discussion of where EM might become a bottleneck. An analysis of scalability (e.g., in terms of latent dimensionality, trial count, or observation length) would strengthen the paper’s claims about practical applicability and robustness. Even a partial exploration of variational alternatives would also help contextualize how “easy to fit” this model remains.

*Kernel function*:
The use of an approximate GP prior over system parameters is presented as the paper’s main technical contribution. However, the empirical evaluation does not really explore how the structure or complexity of the kernel, or the form of the input u, affects performance. In the experiments, the modulators are closely linked to the observed neural dynamics (e.g., categorical or smooth latent variables), making it difficult to assess the generality or necessity of the kernel. I could image that there are forms of inputs that would benefit from different kernels. Since the GP prior induces a specific form of smoothness in the system parameters, it could strengthen the paper to include ablations comparing different modulator structures and kernel variants (e.g., alternative basis functions or GP approximations), especially given the centrality of this mechanism to the paper's novelty. For instance, what prior would you recommend if smoothness assumptions over system parameters are violated (e.g. task switches or abrupt transitions)? How do find an optimal $\kappa$ depending on smoothness?

*Input-driven DS*:
The paper briefly mentions an important conceptual parallel to time-varying or input-driven LDSs, where external inputs u_t​ enter the system via a factor-loading matrix B. This structure is known to induce time-dependent fixed points or more complex attractors (e.g., ring attractors) if the input signal is sufficiently expressive. While not explicitly framed this way, the proposed model can be viewed as implicitly implementing a snapshot attractor. Given the close mathematical similarity to classical input-driven LDSs, it would be helpful to more explicitly articulate what is novel about the kernel-based parameterization (particularly e.g. compared to Input Switched Affine Network that are cited by the authors). At least to my understanding, similar expressive dynamics could potentially be captured by using known u_t​ in combination with a flexible encoder. A systematic empirical comparison to such baselines could help clarify this distinction and further strengthen the case for the proposed approach.

---

> ### Author Rebuttal · Authors · 2025-07-30
>
> We are very grateful to the reviewer for their detailed and insightful comments, and overall positive review. You and other reviewers found merit in the conceptual advance, found the paper well written, and suggested better outlining and testing the scopes of applicability. This provides valuable feedback towards improving our contribution.
>
> Thank you for pointing out the scope of the experiments conducted so far in the paper, which we hope to highlight model recovery and inference on real-world experiments. Your feedback on more challenging experiments targeting the model assumptions is echoed by reviewers and is a valuable concern. To address this overall weakness, we introduce below a new experiment that readily fits within the experiments considered so far while targeting the idealized conditions of low dimensionality and model alignment. Furthermore, we point you to our answer to Question 4 of Reviewer pLGj, which details **new numerical calculations** of the empirical covariance in $x$ given $u$. As per Appendix A.2., this provides a diagnostic feature and proxy for our approximation error to any true nonlinear dynamics in the system.
>
> ## Mechanistic model of ring-attractor dynamics
>
> Inspired by reviewer feedback, we provide an alternate synthetic ring attractor experiment based on the neuroscience literature—see review by Hulse and Jayaraman (2020, “Mechanisms Underlying the Neural Computation of Head Direction”). The intention with this additional task is to test the CLDS and its inference in a synthetic data modality, where we know the underlying computation, but which, importantly, is not generated from a CLDS model, to test the model in mispecified settings.
>
> **Model description**: We write a model of continuous ring attractor dynamics with bumps of activity integrating angular velocity (Zhang, 1996, “Representation of spatial orientation by the intrinsic dynamics of the head-direction cell ensemble: a theory”); a model and computational account of spatial orientation shared across many species. In this model, attractor dynamics are implemented through head-direction preferential units arranged in a “ring”, with short-range excitation and long-range inhibition forming a bump of activity for a specific unit in a specific head-direction, with a velocity integrator causing a shift in the location of the bump as the head-direction changes. Concretely, the dynamics of the new generative model follow
>
> $$ \frac{1}{\tau} \dot u(\phi, t) = - u + f(w * u) + v \frac{\partial}{\partial \phi} u(\phi, t).$$
>
> Here, $u(\phi, t)$ represents the network activity of cells with preferred head-direction $\phi$ at time $t$, $f$ is a (ReLU) nonlinearity, $w$ is a “Mexican-hat” convolution filter, and $v_t = \dot \theta_t$ is the instantaneous difference in head-direction $\theta_t$. For implementation purposes, $u$ is discretized to $u_t \in \mathbb{R}^N$, $t \in \{1, \dots, T\}$, with each unit  $u_i$ having preferred directions $\phi_i$ regularly spanning the interval $[0, 2\pi)$, and $w * u$ is implemented with a circulant matrix $Wu_t$. We further include additive Gaussian random noise to make the process stochastic. We use this activity $u$ as our observations, of dimension $N=32$. The head direction $\theta_t$ still acts as our conditions, sampled the same way as in our previous synthetic HD model.
>
> **Results**: We know the underlying model captures high-dimensional (N=32 here) ring attractor dynamics. When fitting a low-dimensional CLDS model, we expect the latent dynamics to capture the ring-attractor structure and the nonlinearity of the dynamics, with a linear emission model bringing it back to the high-dimensional activity. This is precisely what we found when fitting the CLDS—all necessary figures to relay the results will be included in the final revision. After a search over kernel hyperparameters, we found the best-performing model to encode a ring of fixed points that closely resembles the synthetic HD fixed points, and with eigenvalues closer to the ones observed from the CLDS fits on the mouse ADn recordings. The results make us confident that the CLDS can capture this nonlinear structure even under model mismatch.
>
> ## Response to individual weaknesses
>
> **(1) Spike data**: We agree that the extension to additional noise models (e.g. Poisson) is of interest, and we intend to follow up soon with a full implementation of this. There are no technical hurdles in principle—we just have not prioritized this particular direction. In our experience, simple linear Gaussian methods perform well and are popular among practitioners in neuroscience. For example, Gaussian Process Factor Analysis (GPFA) uses a Gaussian noise model.
>
> We agree with the reviewer that this extension would enhance the paper. However, we think the core conceptual advances of the paper do not rely on the noise model. For example, we think most neuroscientists would regard the head direction system as an inherently nonlinear system. Here, we show that one can model it using linear methods (assuming that head direction is simultaneously measured with neural activity).
>
> **(2) EM Scaling**: For now, we are focused on obtaining point estimates of model parameters &mdash; i.e. the functions $A(u), b(u), C(u), d(u)$. We mentioned MCMC and variational inference as possible extensions for obtaining an approximate posteriors over these parameters/functions. In contrast, EM only provides point estimates (the maximum of the posterior). In other words, this is an apples-to-oranges comparison. Relative to EM, we would expect that MCMC to be much more expensive (since it involves sampling) and variational inference to be moderately more expensive (since it involves an optimization problem with additional parameters).
>
> We think that EM is scalable for problems of interest in neuroscience. Extremely high-dimensional latent spaces would be susceptible to overfitting anyways. Furthermore, it is not clear that alternative methods are more scalable. In particular, any method that computes the marginal log likelihood of $y_1, \dots, y_T$ integrating over $x_1, \dots, x_T$ will perform the Kalman filtering step anyways, which accounts for a large chunk of the computation time in EM. In particular, the M-step is very cheap. In principle, it is possible to do gradient ascent on the marginal log likelihood, but then one must tune learning rates and in general take much smaller steps in parameter space.
>
> In practice, we believe the biggest challenge for scaling will be datasets of very long duration (large $T$). Here, it is possible to leverage parallel computation on a GPU to perform the Kalman filtering, reducing the scaling behavior from $T$ to $\log T$. See Sarkka & Garcia-Fernandez (2020, “Temporal parallelization of Bayesian smoothers”)
>
> **(3) Kernel function**: Draws from a GP with an RBF kernel are infinitely differentiable. A Matern kernel would be a weaker prior (leading to less smooth functions). This would introduce additional hyperparameters and lead to a complex model selection process. For example, the kernel hyperparameters are known to be unidentifiable in a simple regression setting. See Zhang (2004, “Inconsistent estimation and asymptotically equal interpolations in model-based geostatistics”).
>
> A CLDS model can account for abrupt task switches or transitions if they are associated with a measurable behavioral event. For example, we show this in our analysis of the nonhuman primate reaching task by selecting $u_1, \dots, u_T$ to be a step function centered at the time of movement onset.
>
> If abrupt task switches or transitions are unobserved (i.e. latent variables), then one could consider a switching linear dynamical system. We discuss the complementary challenges and advantages of this alternative model in the paper. In particular, inference in this model is less straightforward because Kalman filtering and smoothing is not possible. Speaking from personal experience, we found that open-source implementations of these models were quite difficult to tune in practice.
>
> **(4) Input-driven DS**: The most relevant prior work we are aware of is by Foerster et al. (2017, “Input Switched Affine Networks: An RNN Architecture Designed for Interpretability”) and more recently the SSM models like Mamba (Gu & Dao, 2024, “Mamba: Linear-Time Sequence Modeling with Selective State Spaces”). Neither of these papers explicitly mention attractor dynamics or ring attractors specifically.
>
> The main difference is that these models are applied to natural language processing tasks and therefore take in discrete tokenized inputs. The CLDS model we propose takes inputs over a *continuous* space and furthermore incorporates a smoothness prior over this space. Thus, in the head direction example, our model supports an exact ring attractor.
> We think that an input-switched affine network could in theory learn to approximate a ring attractor if the head direction variable were binned onto a discrete number of values. However, (1) it does not incorporate noise in the latent space, a crucial component of SSMs and probabilistic modeling, and (2) the model would have a hard time learning this from a random initialization because it has no smoothness prior and therefore no inductive bias towards learning similar representations for nearby heading directions.
>
> Thank you again for your time; we look forward to your final assessment.

---

> > ### Comment · Reviewer_hqD4 · 2025-08-05
> >
> > Thank you for the detailed response and for the interesting new experiments. Just to clarify my earlier remark on VI: I'd misunderstood your comments as suggesting VI approach for the E-step (à la sequential VAEs) to jointly approximate the latent trajectories, but given your linear model, this is indeed not really sensible.
> >
> > As mentioned in my original review, I find this paper quite nice and compelling, and given the new results, and echoing the other reviewers’ overall positive resonance, I am happy to vote for acceptance.

---

### Official Review · Reviewer_PoZQ · 2025-06-24

**Clarity:** 4
**Significance:** 3
**Originality:** 3
**Rating:** 5
**Confidence:** 4

**Summary:**

In this paper, the authors introduce a class of latent dynamics models for modeling neural activity. In particular, they propose time-varying linear dynamical systems, where the dynamics and readout parameters vary smoothly as a function of external inputs at each point in time. They demonstrate how these models can be efficiently trained, and show that they are performant on both synthetic and real data tasks.

**Questions:**

- Can the authors comment on the tradeoff between latent dimension $D$  and approximate GP prior truncation length $L$ in terms of expressivity and risk of overfitting?
- One interpretation of the parameterization $\textbf{b}(\textbf{u}_t)$ is that the external inputs undergo a nonlinear featurization before being linearly read into the state updates (setting aside the dynamics matrix also depending on $\textbf{u}_t$ for now). How well would other models do with this nonlinear input featurization step? I would imagine that even a standard LDS model could improve greatly on nonlinear tasks with this modification.

**Ethical Concerns:**

["NO or VERY MINOR ethics concerns only"]

**Final Justification:**

I am keeping my original positive score. There were no major issues raised.

**Limitations:**

yes

**Quality:**

4

**Strengths And Weaknesses:**

Strengths:
- The authors convincingly show that both training and inference of CLDS models can be efficiently done via classical techniques such as Kalman smoothing and EM.
- There are many avenues towards interpretibility of CLDS models (e.g. linearizing around a particular input, or marginalizing out the inputs to directly yield possibly nonlinear composite dynamics)

Weaknesses:
- While the authors do a great job comparing CLDS to models traditionally used in neuroscience (such as (r)SLDS), the class of models introduced here---linear RNNs with parameters being time-varying solely through input dependence---is not at all new in the ML literature, and is a popular formulation of modern selective state space models (e.g. Mamba). The authors should comment on this, and perhaps discuss how the Bayesian perspective (which is indeed distinct from that setting), or the distinct parameterization strategy for achieving input dependence (truncated approximate GP prior as opposed to an MLP), yield unique advantages.

---

> ### Author Rebuttal · Authors · 2025-07-31
>
> We are very grateful to the reviewer for their time and insightful questions. We are pleased to see your overall positive assessment of our work!
>
> ### Weaknesses
>
> The point on modern state-space models is a great one. We had mentioned input-switched affine networks in the first submission, but missed Mamba and its related literature. Our revision will include a longer discussion emphasizing the following:
>
> - **Presence of noise**: SSM models developed in the ML community do not model noise in the latent space. This is thought to be important when analyzing neural data due to limited dataset sizes and noisy observations. Latent noise complicates the inference process and common strategies for inference include EM and marginal optimization for tractable models and variational inference or MCMC for intractable models.
> - **Structure of the state-space**: SSM models developed in the ML community are often applied to natural language processing tasks and therefore take in discrete tokenized inputs. The CLDS model we propose takes inputs over a *continuous* space and furthermore incorporates a smoothness prior over this space, which is an important inductive bias. Thus, in the head direction example, our model supports an exact ring attractor. Furthermore, our work shows how input-selective models can be used to model continuous attractors (e.g. ring attractors for heading direction). Given the fundamental importance of attractor dynamics to modern theories of neural circuit dynamics, we think this is an important conceptual advance which cannot be found in existing ML literature.
> - **Models of nonlinearity**: Finally, we can comment on the GP v.s. MLP modeling. GPs importantly carry a prior over functions, which is of great help in inferring the non-parametric forms of the coefficients in the low-data regimes often seen in neuroscience (as mentioned above). Furthermore, this same prior makes for the potential to obtain full posterior estimates over these coefficients modeled with GPs—we discussed in the paper how this could be done in Lines 127-128.
>
> ### Questions
>
> Addressing your questions one-by-one:
> - **Dimensionality and GP trade-off**: An important aspect of this tradeoff is that the CLDS is *linear* in dynamics ($D$-dimensional) and is nonlinear over conditions (with degree of nonlinearity enforced by $L$). Hence, while both trade off in final expressivity, they do so in complementary ways that model predictions (in time v.s. over different conditions) can help further parse out. Second, on the topic of overfitting, we typically treat $L$ as fixed to a high-enough value to allow sufficient expressivity by the GP prior, as opposed to an optimized hyperparameter—see also our related comments to Reviewer pLGj on this topic. The dimension $D$ is typically variable and searched over, but selected through cross-validation and restricted to be low-dimensional, both avoiding overfitting.
>
> - **Nonlinear biases**: We agree with your intuition here. It is worth noting that the specific model mentioned (standard LDS + nonlinear $\mathbf{b}(\boldsymbol{u}_t)$) is a subcase of the more general CLDS with both $\mathbf{A}(\cdot)$ and $\mathbf{b}(\cdot)$ nonlinear: while the GP prior implicity favors against that, one *could* learn a degenerate set of weights emphasizing the constant function for $\mathbf{A}(\cdot)$, making it constant over $\boldsymbol{u}$. Still on the topic of biases, one can bypass the dynamics entirely by using $\mathbf{d}(\boldsymbol{u}_t)$ to essentially fit a nonlinear tuning curve or PSTH to each neuron (disregarding how single-trial dynamics vary around these trial-average firing rates). We think it is feasible and interesting to see how all of these “ablations” of the CLDS model perform on various experimental datasets, in particular the nonhuman primate reaching dataset.
>
> Thank you again for your time; we look forward to your final assessment.

---

> > ### Comment · Reviewer_PoZQ · 2025-08-04
> >
> > I thank the authors for their answers and clarifications. I would like to keep my score as is.

---

### Official Review · Reviewer_JFre · 2025-06-29

**Clarity:** 3
**Significance:** 2
**Originality:** 3
**Rating:** 5
**Confidence:** 3

**Summary:**

This paper introduces a framework named the CLDS: Conditionally Linear Dynamical Systems, a novel probabilistic modeling framework for the neural population activity along with the task-related or the behavioral-related covariates. CLDS first have the interpretability and tractability of classical linear dynamical systems. And also, it has the flexibility of modern non-linear powerful models by allowing system parameters (e.g., transition and emission matrices) to vary as a function of observed covariates via the pre-defined Gaussian Process (GP) priors. Like most works in this probabilistic modeling genre, the CLDS derives an efficient MAP inference method using approximate GP basis expansions and EM optimization. As for the experiment part, this work has done extensive results on both synthetic and real neural datasets of mouse and macaque. The results manifest that CLDS achieves superior performance aganist the baseline methods, especially in data-limited situations.

**Questions:**

1. What's the most difference and novelty of your probabilistic modeling framework compared to existing works? If there is a table that compare between these methods, that would be apparent to the audience. Also, there is a recent paper [1] whose method that can be compared with the proposed CLDS.
2. How does u_t enables non-linearity and how much could it introduce?
3. Could this framework extend to non-Gaussian likelihoods?

[1] Multi-Region Markovian Gaussian Process: An Efficient Method to Discover Directional Communications Across Multiple Brain Regions. Li, et, al. ICML 2024.

**Ethical Concerns:**

["NO or VERY MINOR ethics concerns only"]

**Final Justification:**

See my full comments below. I maintain my opinion that the manuscript is suitable for publication and keep my score at 5.

**Limitations:**

Please refer to my 'Questions' section.

**Quality:**

3

**Strengths And Weaknesses:**

I think that the strengths of this work lie in its new probabilistic modeling framework with certain extent of locally interpretation. The idea of adding some conditional covairates to the traditional LDS framework is interesting, which lacks some expressive power. With some new derivations, the work also presents the framework with closed-form EM updates. As for the empirical experimental results, the model demonstrates good performance on both synthetic and multiple real-world neural datasets. On the other hand, I think the concerns of this method lies in what's the background neuroscience intuition behind this proposed elegant framework, what part of the method is brain-inspired. How could you prove the soundness of CLDS besides the experimental results.

---

> ### Author Rebuttal · Authors · 2025-07-31
>
> We are grateful to the reviewer for their time and constructive comments. We are pleased to see your overall positive assessment of our work! Below are the answers to your questions.
>
> ### Weaknesses
>
> The CLDS model is a statistical description of condition-dependent high-dimensional time series with latent structure, primarily motivated by and dedicated to the analysis of neural dynamics. We would not consider the CLDS to be “brain-inspired” in the sense of a biophysically realistic circuit model.
>
> Regarding the soundness of the approach besides experimental results, we would point to the fact that the CLDS model is a natural extension of a lot of existing, influential models. In particular, we would point to past work using linear network models of neural circuits (see citations [4-8] in the paper) and more recent work that develops nonlinear dynamical systems models (e.g. citations [9-14] in the paper). Our central argument is that CLDS models provide some of the nice benefits of both.
>
> In practice, neuroscientists are often interested in finding the relationship between experimental task variables or observed behavioral covariates with neural data. Common models (e.g. LFADS) are fit either in an entirely unsupervised manner or incorporate this task and behavioral covariates in a highly complex, nonlinear manner. CLDS models strike a nice balance between interpretability and expressivity.
>
> ### Questions
>
> Answering your questions in order
> 1. There is a very broad variety of statistical models for neural data analysis that are targeted at different specific circumstances. While it is not easy to fit all of these methods into a table, we think the most important advance of our paper is to show that simple and interpretable models (i.e. linear) can be tweaked to model complex (i.e. nonlinear) neural circuit dynamics.
> 2. $\mathbf{A}(\boldsymbol{u}_t)$, $\mathbf{b}(\boldsymbol{u}_t)$, $\mathbf{C}(\boldsymbol{u}_t)$, and $\mathbf{d}(\boldsymbol{u}_t)$ are all nonlinear functions of $\boldsymbol{u}_t$ and belong to the nonparametric family of functions determined by a Gaussian Process (GP). To your question, with the coefficients framed as GPs, they allow for a nonlinear dependency on the conditions, making the system linear in dynamics and *nonlinear* in conditions. The amount of nonlinearity introduced depends on the kernel and its parameters. For example, if we use RBF kernels, the length scale determines the probability of how much high-frequency content (or wiggliness) is in the function. This provides an appropriate mechanism for controlling the level of smoothness either using some prior knowledge or in a data-driven manner.
> 3. Certainly! We comment on how this can be done on lines 163-170 in the submitted manuscript.
>
> Finally, thank you for providing a pointer to this missed reference; we will make sure to incorporate it.
>
> Thank you again for your time; we look forward to your final assessment.

---

> > ### Comment · Reviewer_JFre · 2025-08-05
> >
> > I thank the authors responses and clarifications. I find it interesting in "making the system linear in dynamics and nonlinear in conditions". I will maintain my original score.

---

### Official Review · Reviewer_pLGj · 2025-07-03

**Clarity:** 4
**Significance:** 3
**Originality:** 3
**Rating:** 4
**Confidence:** 4

**Summary:**

The authors propose a novel approach that they term “Conditionally Linear Dynamical system” (CLDS) to model the neural population dynamics underlying activity of large groups of simultaneously recorded neurons. Their method complements a growing set of related methods to fit and interpret neural population dynamics. With CLDS, the authors aim to strike a balance between explanatory power and interpretability. Neural activity is modeled with a linear dynamical system (good interpretability) whose parameters are conditioned on task and/or behavioral variables that are defined by the experimenter, which makes the resulting dynamics overall non-linear (good explanatory power). The authors show analytically how CLDS relates to and differs from several previously proposed methods to fit neural population dynamics. They apply their method to three example datasets. One example consists of simulated responses based on a model consistent with CLDS (a test of model/parameter recovery), the remaining two examples are fits to open experimental datasets from mice and monkeys.

**Questions:**

(1) For a given choice of conditional variables u_k, the definition of the models includes several meta parameters like the dimensionality of the latent space and the parameters of the GP kernel. The authors mention that some of these parameters are optimized with cross-validation, while others appear to be fixed ad-hoc (e.g. the parameter L). Fig. 6 only provides limited insights into this cross-validation procedure for a single dataset. The latent dimensionality D employed in the examples if quite low, 2D (?) in the mice dataset and 5D in the primate dataset. These values seem low compared to past estimates of the dimensionality of neural activity, which in similar primate datasets was often found to be in the range of 10-15 dimensions. One worry is that such a discrepancy may imply that CLDS is more data-hungry than the authors suggest, at least for the employed parameterization of the Gaussian Processes in those examples. More generally, the demonstration of a robust, systematic approach for determining all meta-parameters would seem to be required to make this method broadly usable in practice.

(2) The linear models that are a component of the proposed methods are known to be ill-posed, in that “dynamics” and “input” components (A and b in Eq 1a) can be non-identifiable, i.e. different combinations of inputs and dynamics can explain the observed activity equally well. This issue of non-identifiability is separate to the one between the latent and observation stages discussed by the authors. What is the implication of this non-identifiability for the application and interpretation of CLDS fits? I would guess that the addition of the conditional dependency of the linear dynamics on u_k could create an additional “level” of non-identifiability, in that it is not clear what variables u_k should be included in a model, and how exactly they should be parameterized (see also my comments below). Does this second type of non-identifiability occur in the model? If so, do the authors have a strategy on how to deal with it?

(3) Fig. 2 show the main validation of the method on simulated data. These results are encouraging, but the scope of this validation is quite limited. The authors consider one specific type of 2D latent dynamics that was produced with the CLDS model and ask if the model can retrieve the underlying parameters. However, it is not clear how well this example matches the experimental realities. For example, what happens to the fits when the observation noise is increased? It is currently not clear how if the simulated observation noise matches those required to explain neural data. Also, what happens if there is a mismatch between the structure of the ground truth model (inputs, u_k) and the structure of the fitted model? Can the model reliably retrieve other types of dynamics, including higher dimensional dynamics, rotational dynamics, or non-normal dynamics?

(4) The authors say that they expect CLDS to provide a good estimate of the “composite” non-linear dynamical system when u_k and x_t “tightly” co-determine each other. But how would one decide if the later condition applies? In a setting that is more akin to a switching LDS, with switches determined by some latent variable that is not obviously linked to a specific location in state-space, CLDS would presumably fail to account for the resulting dynamics. Is there some diagnostic feature of the data or the model that could be used to identify such failures?

(5) I was confused by the non-linear dynamics in Fig. 2b. The ground-truth model appears to be a CLDS model, so the dynamics in Fig 2b does not seem to “appear” anywhere in the definition of the ground truth model. In fact, it would be interesting to see how well CLSD can fit data that was generated from a ground-truth model that exactly matches that in Fig. 2b.

(6) The fits of CLDS to the example experimental datasets are also promising, but again the model validation is quite limited. Fig. 3b shows that a few neurons’ dependency on theta is captured, and that for many neurons the mean activity is captured. But what else can be gained from that plot? Likewise, Fig. 3c shows that for a few neurons the tuning curve for theta is well captured. But what about all the other neurons? Dynamics as in Fig 4 are typically shown as condition-averaged trajectories (see for example the original LFADS paper). How well does the model capture those, e.g. compared to LFADS? Also, how exactly was the 3d-space in Fig. 4b defined?

(7) One peculiar choice in the fits of Fig. 4 is the definition of the condition variable z_t, which switches from 0 to 1 100ms after reach onset. Notably, this variable has no immediate parallel in the task, but rather seems to have been chosen by the authors to match the time at which firing rates across the population consistently increase. I find this choice problematic, because it involves a degree of “hand-engineering” the solution found by CLDS, in that it assumes that if a change in dynamics is happening during a trial, it must happen at that (human-chosen) time. I would have found this example more convincing if the authors had chosen time within trial as one of the conditional variables and had then shown that CLDS recovers a change in dynamics (and potentially inputs) sometime after the go cue.

**Ethical Concerns:**

["NO or VERY MINOR ethics concerns only"]

**Final Justification:**

The authors' answers have clarified several of the analyses and findings presented on the simulated and experimental datasets included in the original submission. The authors also seem to have made some effort to more rigorously test the limits of their method and how it may fail in practice, although I find it somewhat difficult to fully assess these novel contributions without seeing the revised manuscript (which is not possible based on current Neurips rules). I believe such tests will be critical to the ultimate success of this method. Nonetheless, based on the clarifications and improvements shown in the responses I have increased my rating from 3 to 4.

**Limitations:**

See questions above.

**Paper Formatting Concerns:**

no concerns

**Quality:**

3

**Strengths And Weaknesses:**

The presented methods (including analytical derivations of expectation-maximization steps to fit the model in several settings) are described in detail and could be an interesting, original addition to current methods. But while the proposed method seems promising, the results appear quite preliminary. The three example applications are limited in scope and are somewhat insufficient to fully assess the value of the method. As I explain in more detail below, several important controls are missing and in general the limits of the models and fitting procedures are not sufficiently explored and discussed.

---

> ### Author Rebuttal · Authors · 2025-07-30
>
> We are very grateful to the reviewer for their detailed and insightful comments. We address your questions point by point below, and point to our reply to Reviewer hqD4 (due to space constraints) which details a "Mechanistic model of ring-attractor dynamics" inspired by Question #5 in your review.
>
> ## Responses to questions
>
> **(1) Hyperparameters:** Tuning hyperparameters is a common challenge across many models, with cross-validation being the *de facto* approach. Although CLDS models are a relatively simple class of models (e.g., in comparison to any deep network based approach), a comprehensive grid search of all potential hyperparameter settings is still computationally infeasible. Here, we stick to a relatively simple choice of an approximate GP kernel with a single hyperparameter governing smoothness (i.e. the lengthscale). We tune this by cross-validated co-smoothing.
>
> While it is possible to treat $L$ as a hyperparameter tuned to improve performance, we propose to instead set $L$ to a large enough value that results in negligible error in the GP kernel approximation. As $L \to \infty$, we pay a larger computational price in fitting the model (because we have more basis functions), but the statistical properties of the model are essentially unchanged because high-frequency basis functions have nearly zero amplitude. This idea is well established in GP regression literature; see for example (Greengard et al. 2025; "Equispaced Fourier representations for efficient Gaussian process regression from a billion data points."). Hopefully, these clarifications make it clear that $L$ was not selected in an “ad hoc” manner, but set to a “large enough” value. We will explain this in more detail in the revision, as well as numerical error metrics.
>
> Regarding dimensionality, we are unaware of conclusive evidence that 10-15 dimensions is optimal to model nonhuman primate motor cortex. Could the reviewer give us a more concrete citation? In our hands, we find that we get substantially diminishing returns after fitting a ~5D model whether the model is a classic LDS or a CLDS. We report in **Table 1** below co-smoothing results as we take $D \in \\{3, 5, 10, 15\\}$.  In our revision, we will include a more comprehensive sweep over this hyperparameter as a supplementary figure.
>
> | **D** | 3 | 5 | 10 | 15 |
> | ---| ---| --- | --- | --- |
> |**CLDS** | 0.229 | 0.232 | 0.207 | 0.168|
> |**LDS** | 0.211 | 0.193 | 0.196 | 0.17 |
>
> **Table 1**: Co-smoothing reconstruction of single held-out neurons from the test-set, over varying latent dimensionality $D$, averaged over two random seeds.
>
> **(2) Identifiability:** Our revision will clarify the set of equivalent CLDS model solutions, known from the literature on LDS models (Glover & Willems, 1974, “Parametrizations of linear dynamical systems: Canonical forms and identifiability”). We note that this form of non-identifiability is relatively mild compared with many alternative approaches. Anything nonlinear dynamical systems model or deep neural network approach (e.g. LFADS) will have much more extreme forms of non-identifiability.
>
> Although the latent space of CLDS is only identifiable up to these transforms, other quantities are uniquely identified. First, eigenvalues of the dynamics matrices are also identifiable (and thus the dynamical regimes), which we make sure to convey and asses recovery on the two head-direction tasks. Second, for a fixed set of inputs $u_1, \dots , u_T$ and a fixed estimate of CLDS model parameters the conditional distribution of observations $y_1, \dots, y_T$ is an $NT$-dimensional Gaussian with identifiable mean and covariance. The parameters of this Gaussian are efficiently computable via Kalman smoothing. Furthermore, these parameters have scientific value. For example, the covariance characterizes across-neuron and across-time noise correlations. See, for example, Panzeri et al. (2022, "The structures and functions of correlations in neural population codes.").
>
> **(3) Nonlinear experiment:** Thank you for this insightful perspective, it highlights an opportunity for improving the soundness of our approach. Our motivation behind Figure 2 was to show how a canonical nonlinear neural circuit (a ring attractor) could be instantiated by a conditionally linear model. We show (not surprisingly) that we can recover the parameters of this model when fit to noisy simulated observations. Now, as prompted by you and Reviewer hqD4, we sought to challenge the inference and **devised a new task**, specifically chosen to reflect well-established high-dimensional and non-normal (chain-like) dynamics of interest in the neuroscience community. The ground-truth model is not a CLDS, providing a model-mismatch experiment. We found that the CLDS was able to reliably uncover the ring-attractor structure of the true model, further attesting to its ability to recover the nonlinear structure we've seen in the mouse head-direction data.
>
> Finally, like any statistical model, CLDS models will fail to provide good fits in regimes where the data is too noisy or scarce. Data requirements will grow with the complexity and dimensionality of the underlying system. Unlike nonlinear/deep network alternatives, linear systems are amenable to a theoretical analysis (although this remains to be an open area of research). See, for example, Hardt et al. (2018) "Gradient descent learns linear dynamical systems", which shows that no local minima exist under certain conditions. Some sample complexity results are also available. See, Tsiamis et al. (2023) “Statistical learning theory for control: A finite-sample perspective”.
>
> **(4) u-x diagnostic metric:** Thank you for this important question; it is challenging but essential to get a good grasp of this relationship, given the importance of inputs in our modeling approach. Already present in the manuscript, we’ve derived in Appendix A.2. bounds to our approximation error for both the CLDS approximation to a “true” nonlinear system (eq. 23) and the composite dynamics to a “true” autonomous system (eq. 42). Both of these bounds revolve around the second moments of the conditions distributions $p(x | u)$ and $p(u | x)$, namely $\xi :=\mathbb{E}_{u}[\mathrm{Tr}[\mathrm{Cov}[x | u_t]]$. We are pleased to **provide numerical estimates of this quantity** in **Table 2** below, computed empirically from posterior estimates in a manner akin to the composite dynamics detailed in the text. A control is obtained by performing the same procedure on conditions shuffled over trials/batches.
>
> | | Fit | Control |
> | ---| ---| --- |
> |Synthetic HD | 5.7 | 13.3|
> | Mouse HD | 8.4 | 30.7|
> | Macaque reaching | 6.4 | 7.6 |
>
> **Table 2**: empirical estimates of $\xi$. The more co-dependent these variables are, the smaller $\xi$ will be, and the better our CLDS approximation is to a ground-truth nonlinear system.
>
> Their relative value and difference to the control help provide this diagnostic feature. These quantities will be included in the updated manuscript, and the methods along with the open-source code.  Finally, regarding the SLDS example, we refer to points (1) and (7) on general model selection.
>
> **(5) CLDS decompose nonlinear dynamics** It is worth noting that Fig. 2b represents the ground truth composite dynamics, not the ground-truth model itself. Nonetheless, and directly to your point, we have developed above an additional example where we fit a CLDS to a mechanistic ring-attractor model (see response to hqD4). We hope that our description and the pending figure will satisfy your questions.
>
> Part of our motivation here was to show that a CLDS model can, in fact, be hand-constructed to produce simulated ring attractor dynamics. We believe this approach of constructing nonlinear dynamics from linear dynamics per condition is a conceptual point worth making to the neural modeling community.
>
> **(6)** Regarding the head direction system, we will include a supplemental figure showing all neural tuning curves (all of which are well-fit by the model). Regarding condition-averaged trajectories, thank you for pointing this out, our revision will include them—they look similar for the LFADS and CLDS models.
>
> Importantly, both the tuning curves and condition-averaged trajectories are relatively simple to fit. The hard problem is predicting the statistics of single-trial dynamics and an underlying flow field that is capable of forecasting or predicting heldout activity. This is quantified by the co-smoothing analysis.
>
> **(7) Choice of inputs:** Firing rates of motor cortical neurons rapidly rise at movement onset, see Kaufman et al. (2016, “The Largest Response Component in the Motor Cortex Reflects Movement Timing but Not Movement Type”). In principle, we could have set the step function to coincide with the movement onset of the hand, a standard way to align spike times in reaching tasks, but unfortunately, the data we’re working with didn’t have this information.
>
> The CLDS model does involve a degree of “hand-engineering.” As we explain in the paper, we view this as both a strength and a limitation of the model. It is a limitation because we might miss out on important latent variables by taking a more supervised approach. But, on the other hand, purely unsupervised methods are difficult to fit. Our view is that there is a lack of tools that are easy-to-use and allow users to incorporate domain knowledge into their model. CLDS models fill this gap in the literature.
>
> It is true that the inputs to the model can be engineered in a variety of ways, and therefore fair to ask, “How should one choose $u$”? We can’t give a general answer to this question— the answer will depend on the specifics of each experiment. If a practitioner identifies several possible choices for a particular experiment, we suggest that cross-validation could be used to select between these alternatives.
>
> Thank you again for your time; we look forward to your final assessment.

---

> > ### Comment · Reviewer_pLGj · 2025-08-04
> > **Follow-up questions**
> >
> > I thank the authors for their detailed reply and additional experiments/analyses, which have clarified many of my questions.
> > I still have a few questions:
> >
> > (3) I appreciate the inclusion of an additional non-linear model in the revised manuscript. Regarding the issue of noise (specifically observation noise), I understand that, when activity is very noisy, more data will be required. However, my questions on this point were meant to be more specific: (1) is the amount of observation noise (and the amount of data) in the simulations matched to those encountered in typical neural recordings (like those included in the manuscript)? (2) when trial number is limited, how do the fits “fail” as noise is increased? Are low-variance dimensions simply ignored, but dynamics along high-variance dimensions correctly estimated? Or are estimates along all dimensions (e.g. eigenvalues of the dynamics) affected by increased noise?
> >
> > (7) Thank you for the explanation. I still would like to understand though if setting u_k = t (as the authors themselves suggest) would be feasible for this dataset. If so, did the authors try to fit such model?

---

> > > ### Author Response · Authors · 2025-08-07
> > >
> > > Thank you for following up on your remaining questions.
> > >
> > > (3) We are happy to provide some new results on the impact of noise on our quality of fit. In the table below, we report the results of our inference method on the first synthetic ring-attractor experiment presented in the paper for a varying emission noise scale, keeping all other experimental setup parameters exactly the same as in the paper. We see that even as we reach a co-smoothing value of 0.21, which is lower than our fit on the monkey data, we still have decent $\mathbf{A}$ recovery and near-perfect $R$ recovery.
> > >
> > > True noise log scale | -2 | -1 | 0 | 1|
> > > | --- | --- | --- | --- | --- |
> > > Recovered $R$ log scale  | -1.97 | -0.98 | 0.02 | 1.02|
> > > $\mathbf{A}(\cdot)$ recovery error  |0.01 | 0.02 | 0.11 | 0.32|
> > > Co-smoothing $R^2$ | 0.99 | 0.94 | 0.68 | 0.21|
> > >
> > > **Table 3**: The recovered $R$ scale is the square-root of the matrix 2-norm of $R$. The $\mathbf{A}(\cdot)$ recovery error is computed as the average in 2-norm error (between recovered and true) in eigenvalues over a grid of 50 angular conditions $\theta$. The Co-smoothing $R^2$ is the average validation set $R^2$ on top-5-variance neurons, exactly like in the paper.
> > >
> > > We will include the results as plots in the revision. A full investigation of the impact of noise on model quality is interesting, but outside of the scope of the current paper. However, we view the possibility of conducting such an investigation to be a strength of our model class. Indeed, it would be much more challenging to the point of being hopelessly impractical to develop analogous mathematical theory for deep-network based models. In those cases, one would have to resort to ad hoc simulations and numerical observations.
> > >
> > > To the point of variance, in CLDS models, we expect that signal dimensions with high variance (i.e. columns of $\mathbf{C}(\boldsymbol{u})$ that have large norm) will be learned more accurately and with less data than low signal-to-noise dimensions, similar to what one can show happens for probabilistic PCA. Again, we expect that linear dynamical systems models with Gaussian noise are tractable enough that this could be studied mathematically, and we believe this is a selling point for working with simple, but expressive models. However, we are unaware of existing publications that tackle this precise question.
> > >
> > > (7) Yes, we tried setting $\boldsymbol{u}_t = t$ and found that the model performs reasonably well after tuning the lengthscale. However, the model proposed, where $\boldsymbol{u}_t$ is a step function aligned with movement onset, performs well (if not better). We chose to highlight the step-condition model to make the connection to switching linear dynamical systems more concrete, and to (intentionally) illustrate how we can engineer conditioning variables using knowledge of the data. Finally, the model with a step function has only two fixed points to visualize per condition, whereas the model with  $\boldsymbol{u}_t = t$ has a continuous trajectory of fixed points.
> > >
> > > We hope these points address your final questions, which, in combination with our clarification of your other questions, will suffice for you to reconsider your score.  Thank you again for your time, and we look forward to your final assessment.

---

### Decision · Program_Chairs · 2025-09-17

**Decision:**

Accept (poster)

**Comment:**

In this paper, the authors introduce a nice novel idea of recasting nonlinear neuronal dynamics in terms of conditionally linear dynamics, given behavioral and task variables. This yields highly interpretable and tractable models for analyzing dynamics in neural systems, without sacrificing the nonlinear nature of neuronal dynamics.

With the initial submission there were considerable concerns, brought up by two referees, about the rather limited empirical evaluation, lacking important controls and raising issues about data requirements and model scaling. Other questions concerned the determination of the model’s hyperparameters, model identifiability, and some model diagnostics. These were mostly addressed by the authors during the rebuttal through additional numerical checks, detailed formal arguments, and by adding another evaluation on a ring attractor model. Ref. hqD4 also brought up a very valid point in my mind about the model’s similarity to other input-driven DS/ SSM formulations, which casts a bit of doubt on the novelty of the authors’ approach and should be discussed in the paper’s final version.

Three of the four referees clearly voted for acceptance, while one referee initially remained a bit doubtful, but stated that most of their concerns have been properly addressed and leaned toward acceptance in the end as well. I go with the overall positive assessment and recommend acceptance. I expect the authors to include the clarifications provided in the rebuttal and discussion, and in particular the new evaluation setting, in their final revision!